

# Invited perspectives: Thunderstorm Intensification from Mountains to Plains

Jannick Fischer[3], Pieter Groenemeijer[1,2,*], Alois Holzer[1,2,*], Monika Feldmann[6,*], Katharina Schröer[5,*], Francesco Battaglioli[2,*], Lisa Schielicke[4,8,*], Tomáš Púčik[1,*], Christoph Gatzen[2,*], Bogdan Antonescu[2,7,*], and TIM Partners[9]

[1]European Severe Storms Laboratory - Science & Training, Wiener Neustadt, Austria
[2]European Severe Storms Laboratory, Wessling, Germany
[3]Karlsruhe Institute of Technology (KIT), Karlsruhe, Germany
[6]Institute of Geography - Oeschger Centre for Climate Change Research, University of Bern
[4]Institute for Geosciences, University of Bonn
[5]Department of Environment and Natural Resources, University of Freiburg
[7]Faculty of Physics, University of Bucharest, Romania
[8]Department of Physics and Astronomy, The University of Western Ontario, London, Canada
[*]These authors contributed equally to this work.
[9]See extended co-author list from partner institutions at the end of this manuscript

**Correspondence:** Jannick Fischer (jannick.fischer@kit.edu)

**Abstract.** Severe thunderstorms are among the most damaging and impactful weather phenomena. In Europe, notable clusters occur in the vicinity of complex terrain. These areas not only experience frequent thunderstorms but also show a strong climate change signal with an increasing storm frequency. Despite the relevance of the subject, our understanding of severe convection in complex terrain, particularly in a changing climate, remains incomplete. This review presents the current state
of the research on thunderstorms in complex orography, covering storm severity, modification of pre-storm environments, convection initiation, storm-scale interactions with complex terrain, impactful hazards, numerical modeling and forecasting, climatologies and climate change signals, as well as innovative storm observations. Highlighting the gaps in our understanding, this review underscores the need for a coordinated European field campaign on Thunderstorm Intensification from Mountains to Plains (TIM). Initial plans for the TIM campaign built by participating authors and institutions of this article are briefly
outlined. Obtaining coordinated and dense data on orographically driven storms is a key step toward improving warnings, forecasts, future climate projections, and adaptation measures.

## 1   Introduction

Severe convective storms (SCS) with flash floods, hail, severe wind gusts, lightning, and tornadoes cause a significant and increasing amount of damage across Europe and other mid-latitude regions (e.g., Banerjee et al., 2024). In 2023 alone, a new
record hail size was set for Europe with 19 cm (Eisenbach, 2023) while in the US the insured loss from severe convective storms reached 60 billion USD, thereby also setting a new record and dominating the global total loss (Bowen et al., 2024).



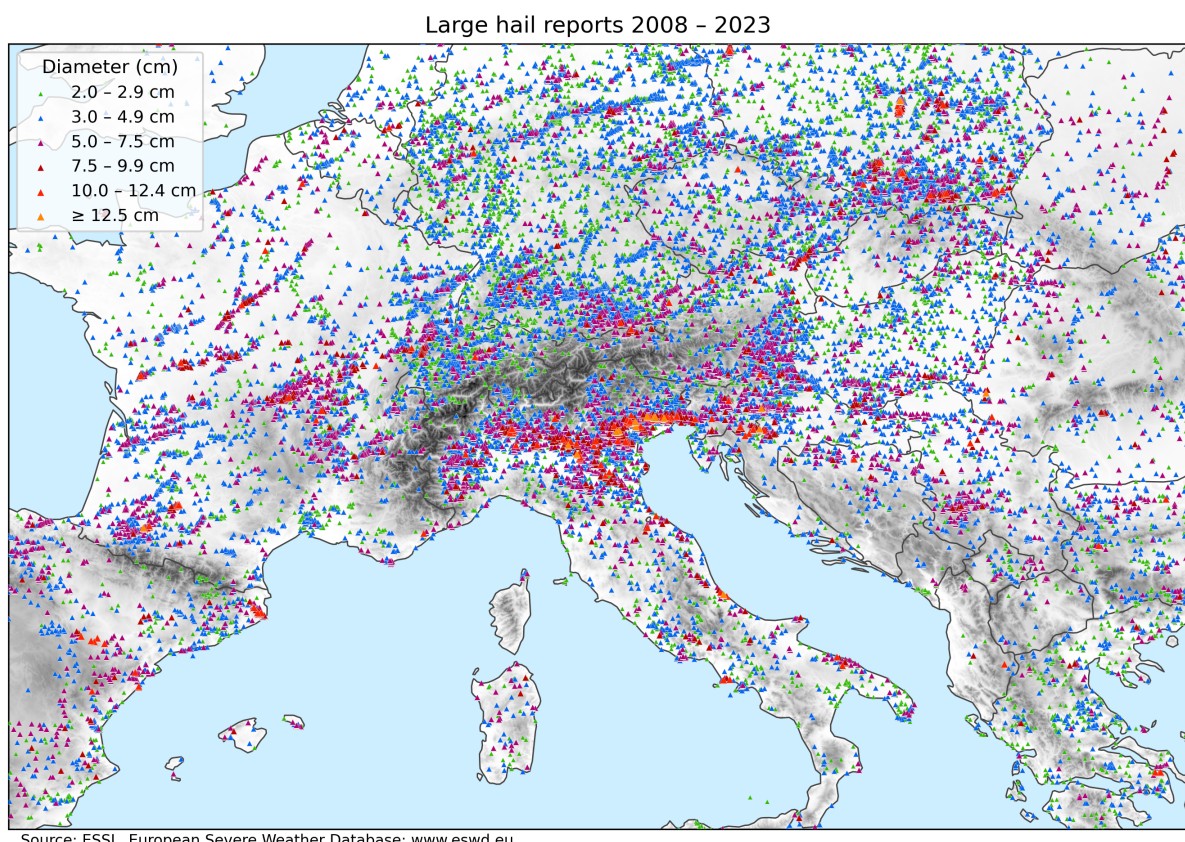

Source: ESSL, European Severe Weather Database: www.eswd.eu

**Figure 1.** Severe hail reports between 2008-2023 in the European Severe Weather Database (ESWD). The topography is qualitatively underlayed in gray with darker shading indicating higher terrain. For more topographic detail and the locations of partner institutions see Fig. 7.

These facts highlight the importance of research on all aspects of severe convective storms to better understand where and why these hazards occur.

Reports of severe weather in the European Severe Weather Database (Dotzek et al., 2009, accessed 10 May 2024) suggest

that particularly severe convective storms are more frequent over the flanking slopes of mountain ranges and the plains that straddle them rather than directly over the mountains. As an example, Fig. 1 shows the clustering of severe hail reports north and south of the Alps and predominantly on the eastern or northeastern flanks of lower mountain ranges. This pattern is supported by indirect but spatially uniform measures of storm severity such as overshooting storm tops (Punge et al., 2017; Giordani et al., 2023), lightning frequency (e.g., Manzato et al., 2022b), and radar-based climatologies in individual countries (e.g.,

Kaltenboeck and Steinheimer, 2015; Punge and Kunz, 2016; Wapler, 2021; Schröer et al., 2022; Feldmann et al., 2023). The same geographic regions include highly-vulnerable population centers and are projected to experience the strongest increases in severe weather occurrence as a result of global warming (Púčik et al., 2017; Rädler et al., 2019; Battaglioli et al., 2023).



**Figure 2.** Questions from the White Paper survey conducted for this article with the European partner institutions listed further below.

Thus, an important question arises: Why are storms in the vicinity of mountain ranges more severe? As reviewed in section 2.1, there are several possible explanations in the literature. However, our understanding of thunderstorm dynamics and mesoscale

atmospheric processes is still incomplete. One reason is that observational datasets to validate existing theories in different regions and scenarios are scarce and, where available, distributed across many institutions and data archives.

This lack of understanding and of shared data is echoed throughout the research field of severe convective storms. As basis for this article, the European Severe Storms Laboratory (ESSL) conducted a survey with institutions in the European atmospheric science community, asking for their perspectives on the most crucial research topics on severe convective storms (Fig. 2). The

contributors to this survey also took part in the writing of this article and are listed in the extended Co-author section. The survey outcome was a pool of over 40 research questions, which are sorted by topic and discussed in detail in section 2. A recurring theme in the responses is that a better understanding of the physical processes controlling severe convective storms and their impacts is needed. Public authorities, research institutions, the aviation industry, and the reinsurance sector share the need to better estimate the frequency, magnitude, and predictability of severe weather.

Presently, a wealth of new sensors and data types is emerging to cater to these needs, including next-generation high-resolution satellite data from the Meteosat Third Generation programme (Holmlund et al., 2021), numerical modeling within the DestinE project (Wedi et al., 2022) and the proliferation of polarimetric Doppler radars across Europe (Saltikoff et al., 2019). Validation of these datasets is key, and requires that they are shared among scientists with the least possible restrictions. This validation is especially important near mountain ranges, not only because of the high severe weather frequency (Fig. 1),

but also because the lack of observations of basic meteorological parameters in the boundary layer and in complex terrain[1] is critical to advance forecasting and nowcasting skill (Bojinski et al., 2023).

The prominent need for shared data and research on severe storms, especially near mountains, can partly be explained by the prevailing focus of past field campaigns either on thunderstorm dynamics away from mountain ranges *or* on terrain influences on the atmosphere but not on the scale of individual severe storms. For instance, several large field campaigns in the United

---

[1]The terms "complex terrain" and "orographic" will be used in this article when referring to the vicinity of mountains.





States, such as VORTEX1/2/SE (Rasmussen et al., 1994; Wurman et al., 2012), PECAN (Geerts et al., 2017), TORUS (Houston et al., 2020), PERILS (Kosiba et al., 2024) and TRACER (Jensen et al. 2024) led to advancements in our understanding of thunderstorm dynamics, microphysics, and processes leading to severe weather. Meanwhile, convective-season field campaigns in the Alpine region, such as MAP (Bougeault et al., 2001), COPS (Wulfmeyer et al., 2011), HyMeX (Ducrocq et al., 2016) and, currently, TEAMx (Rotach et al., 2022), greatly improved numerical models and process understanding with respect to

large-scale orographic flow, boundary-layer exchange processes, autumn convective floods, and processes occurring prior to convection initiation (CI), but much less so regarding the dynamics of individual thunderstorms (e.g., Rotunno and Houze, 2007). Exceptions are RELAMPAGO and its sister campaign CACTI, which took place in Western Argentina where some of the most intense storms worldwide are heavily influenced by the Andes and Sierras de Cordoba mountains (Nesbitt et al., 2021; Varble et al., 2021). However, the setting of RELAMPAGO in the lee of a long meridional mountain range and the

Amazon Basin as a moisture source differs greatly from Europe and other severe storm regions of the world with respect to larger-scale environments, geography, and climatology (e.g., Zhou et al., 2021). Furthermore, South America does not have the wide spectrum of existing instrument networks, institutions, and research possibilities that could be used in central Europe.

Thus, a field campaign investigating individual thunderstorm dynamics near complex terrain in the densely-observable regions over Europe seems overdue, particularly given the potentially escalating storm activity attributed to climate change in

densely populated regions (Battaglioli et al., 2023). To this end, ESSL is coordinating an effort for an international field campaign in Europe under the name TIM (Thunderstorm Intensification from Mountains to plains). In this article, we summarize the most pressing research questions on the basis of an overview of recent scientific studies. The paper is organized as follows. Section 2 outlines key research topics of TIM. Section 3 then provides a synergy for how these topics can be combined in one field campaign and briefly outlines the preliminary campaign strategy, including focus regions and instrument setups.

## 2  Research topics

Our survey within the community has revealed a large pool of research topics regarding severe storms near mountain ranges. These topics are condensed in the following subsections. Section 2.1 starts with an overview of the physical processes important for thunderstorm intensification near mountains. The following sections describe the related research topics more in depth.

### 2.1  Increased storm severity near high terrain

Severe convective storms tend to be frequent near European mountain ranges as shown by the clustering of severe weather reports (see Fig. 1), and underscored by the documented cases of impactful supercells that tracked through such regions (Kaňák et al., 2007; Kunz et al., 2018; Trefalt et al., 2018; Šinger and Púčik, 2020; Wilhelm et al., 2021; Kopp et al., 2023). That said, it is difficult to disentangle to what extent high storm frequency and stronger average intensity are each driving these near-mountain severe weather maxima.

Thunderstorm frequency is largely related to convection initiation (CI) or the lack thereof. Mesoscale orographic processes dominate CI mechanisms in mountainous regions, in contrast to synoptic-scale processes, such as fronts, which are more





important over flat terrain (e.g., Pacey et al., 2023). As illustrated in Figure 3 and discussed in more detail in section 2.3, orographic CI can occur where terrain-induced flows converge and cause deep continued lift in the atmospheric boundary layer (e.g., Kirshbaum et al., 2018), especially if these flows transport and vertically mix boundary-layer moisture and thereby
decrease convective inhibition (CIN) and dry air entrainment near the convergence zones (e.g., Scheffknecht et al., 2017; Serafin et al., 2020; Marquis et al., 2021; Nelson et al., 2022; Göbel et al., 2023). Hence, mountain ranges strongly influence where thunderstorms form (e.g. Nisi et al., 2018; Manzato et al., 2022b). Although severe storms can move quite long distances after CI (e.g., Scheffknecht et al., 2017; Kunz et al., 2018), CI-prone regions tend to have a much higher frequency and coverage of severe storms (e.g., Feldmann et al., 2023; Kvak et al., 2023). Nevertheless, for Switzerland, Feldmann et al.
(2023) demonstrated that while the regions of highest storm frequency and highest storm intensity strongly overlap, they are not identical. Thus, CI is likely an important but not the only factor.

Severe storm intensity is largely determined by the vertical distribution of temperature, moisture and horizontal winds. Most reports of severe weather occur where certain atmospheric "ingredients" (e.g., Doswell et al., 1996) are met, which can be boiled down to (1) high convective available potential energy (CAPE) which drives deep convective updrafts and which is
found where tropospheric temperature lapse rates and low-level moisture are large, and (2) strong vertical change of wind speed and/or wind direction, i.e., vertical wind shear (e.g., Thompson et al., 2003; Taszarek et al., 2020a). Given CI in regions where these ingredients are met, storms tend to become more intense, more organized, and longer-lived, often forming mesoscale convective systems (MCS) or supercells (e.g., Markowski and Richardson, 2010). Thus, one likely reason for the accumulation of severe reports near orography (Fig. 1) is that these ingredients are enhanced there. Several mechanisms exist in that regard,
which are summarized in the rest of this section. Figure 3 illustrates the respective key processes.

Local terrain-induced flow systems such as thermal slope winds and Alpine Pumping (Ćurić and Janc, 2012; Soderholm et al., 2014; Trefalt et al., 2018; Mulholland et al., 2020; Trapp et al., 2020; Katona and Markowski, 2021), wind channeling (Whiteman and Doran, 1993; Hannesen et al., 1998; Ćurić et al., 2007; Geerts et al., 2009; Tang et al., 2016; Feldmann et al., 2024), or lee cyclogenesis (e.g., McGinley, 1982) can cause perturbations in the atmospheric boundary-layer winds. This can
increase the vertical wind shear and thereby severe storm potential. Furthermore, the wind profile influences storm structure and propagation (e.g., Houze et al., 1993). Terrain flows depend on the thermodynamic stability of the atmosphere (e.g., Houze, 2012), but also the terrain shape and dimensions, and the surface use such as cities or cropland. As a result, typical flow systems seem to exist for individual mountain ranges (e.g., Dotzek, 2001; Kunz and Pusskeiler, 2010), but more universal conceptual models for what processes dominate in what situation and location could be not developed thus far.

Another factor is the spatial distribution of low-level moisture across complex terrain. Moisture is often relatively low directly over mountain ridges, leading to less CAPE compared to the surrounding slopes and valleys. Local increase in water vapor content can be found along convergence lines, near lakes, or other regions where moist air can be transported or accumulated, which would lead to larger CAPE, all else being equal. Processes that favor this are large-scale, up-slope moisture flux (e.g., from the Mediterranean) or evapotranspiration from local water bodies or vegetation (Laiti et al., 2014; Feldmann et al., 2024).
In the same vein, the distribution of water vapor in complex terrain is often spatially heterogeneous, which can play a role in the formation of thunderstorms (Adler et al., 2016; Marquis et al., 2021; Nelson et al., 2021; Calbet et al., 2022). Research



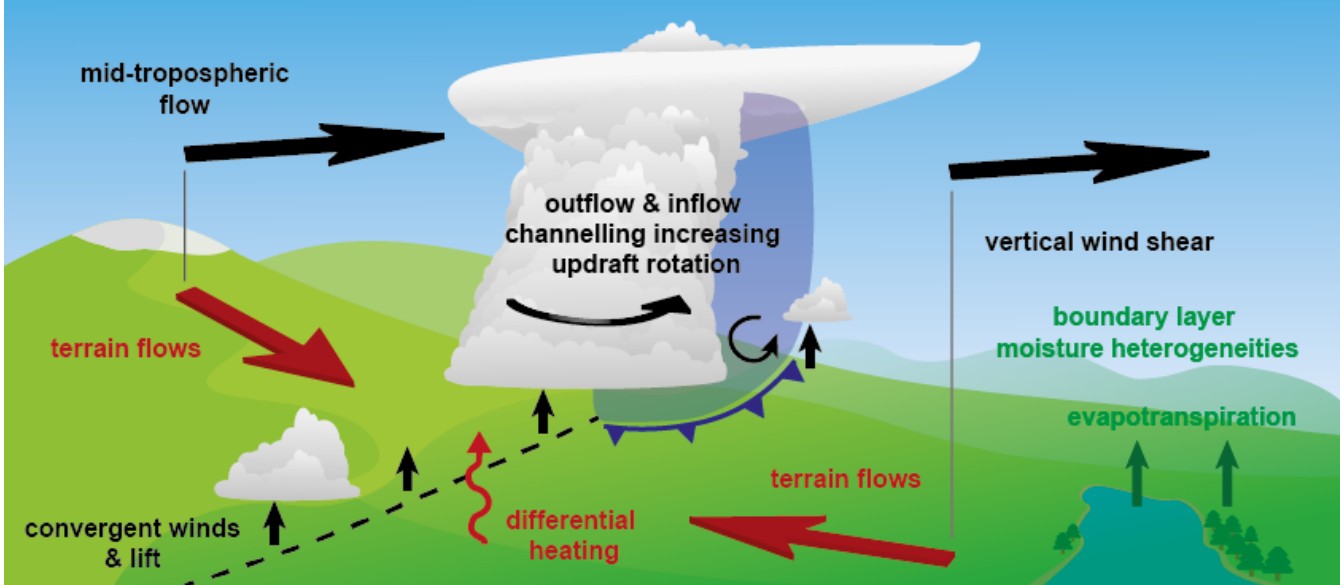

**Figure 3.** Conceptual illustration of processes influencing severe convective storms near and over complex terrain as explained in the text. The term "terrain flows" encompasses all possible terrain-induced near-ground flow perturbations, such as thermally-forced circulation caused by differential heating, mechanically forced ascent or channeling, upstream blocking, and lee-side convergence (e.g., Kirshbaum et al., 2018).

is needed to analyze if storm severity could also be linked to these heterogeneities and to derive remote-sensing retrievals of water vapor that could be used in weather forecasts.

The interpretation of the thermodynamic profile near complex terrain is somewhat complicated by the local modifications
from heat and moisture fluxes. Nevertheless, studies have suggested that with increasing elevation near-surface air parcels that enter a thunderstorm cloud originate higher relative to the background thermodynamic profile that is advected over the terrain, i.e., inflow air does not originate near the surface. At intermediate elevation, this can lead to an increase in convective energy because the most unstable parcels are being lifted (Markowski and Dotzek, 2011; Scheffknecht et al., 2017; Katona and Markowski, 2021; Feldmann et al., 2024). At increasingly higher elevation or on the leeward side (with respect to the deep-
layer winds) of a mountain ridge CAPE is typically diminished because lifted parcels from higher in the atmosphere are drier and the depth of the CAPE layer decreases (Markowski and Dotzek, 2011; Mulholland et al., 2019; Katona and Markowski, 2021; Feldmann et al., 2024). However, all these studies have only considered specific mountain ranges or idealized scenarios and hence it remains unclear how significant these effects are in different geographic regions. For instance, upslope moisture flux on the leeward side of a mountain range may lead to CAPE actually being larger than on the windward side. In this or
similar ways, the geographic location and terrain dimensions likely determine the typical scenario at an individual location.

The above-mentioned convergence regions from terrain-induced flows may also overlap with moisture gradients and take characteristics of dryline-like boundaries (e.g., Pistotnik et al., 2011; LeBel et al., 2021). Such boundaries are known to have an impact on storm severity, mainly through locally enhanced low-level vertical wind shear, moisture, and lift (e.g., Maddox





et al., 1980). However, mesoscale boundaries have not been well documented in Europe although case studies show they can
play a key role in the development of severe weather near mountains (e.g., De Martin et al., 2023; Komjáti et al., 2023).

Lastly, complex terrain also influences the internal structure and processes within existing thunderstorms. This can include channeling of the outflow and inflow of a thunderstorm (Fig. 3), which in turn can feed back to storm intensification via increased storm-relative winds or baroclinic generation of horizontal vorticity (e.g., Feldmann et al., 2024). As discussed in section 2.4, these storm-scale influences are complex and not well-researched. Hence, much research in the storm scale is
needed to better understand them, in addition to the larger-scale, environmental influences in the previous paragraphs.

As suggested by Feldmann et al. (2024), it is the interplay of the factors described in this section that may cause severe storm potential to be enhanced over the slopes of mountain ranges compared to flat terrain or directly over the mountains (Fig. 1). Even slight enhancements are especially relevant in regions like Europe where severe storm potential is often marginal (e.g., Taszarek et al., 2020b). However, what processes are typically at play in which weather situation and in which sub-regions
remains largely unclear and is considered one research focus of TIM.

## 2.2 Orographic modification of the mesoscale environment

The influence of orography on the atmosphere and parameters like CAPE and wind shear opens up several follow-up questions. What influence does the prevailing mid-tropospheric flow direction and speed have on the modification of convective parameters? Why are there regions along the Alps and other mountain chains much more prone to severe storms than others (Fig. 1)?
What are preferential paths for severe storms? At which elevations are environments most or least favorably modified? How significant is the impact of vegetation, water bodies, snow cover, or urban areas?

Related to these topics is also how storms behave when either crossing or moving away from mountains and hence encounter either abrupt or continuous changes in the mesoscale environment. Situations have been documented for both decay or intensification in these scenarios (e.g., Scheffknecht et al., 2017; Trapp et al., 2020; Klaus et al., 2023; McKeown et al., 2024), but
more research is necessary. By documenting exemplary cases within a field campaign such as TIM, detailed observations could help our understanding of these opposing changes.

Furthermore, it is well-established that the characteristic environments for different hazards are statistically different, e.g., between tornadoes and hail (Púčik et al., 2015; Nixon and Allen, 2022). Similarly, Zhou et al. (2021) showed that statistically different hailstorm environments exist, with one of the main differentiating factors being terrain elevation. Hence, the impact
of orographic modifications is likely hazard-dependent. Several studies have shown that channel winds in relatively narrow valleys improve the conditions for supercell tornadoes even if the background environment is otherwise unfavorable (Fujita, 1989; Dotzek, 2001; Bosart et al., 2006; Tang et al., 2016). Whether there are similar mechanisms specifically for hail or other hazards, as is suggested by observations (Fig. 1; Feldmann et al., 2023), is relatively less researched. Targeted observations and measurements as well as simulation studies (e.g., Katona et al., 2016) over different Alpine sub-regions as part of the TIM
campaign could provide deeper insights into the processes of orographic modification.



## 2.3 Convection initiation

Due to its importance for thunderstorm evolution, CI has been a focus of past research. The reader is referred to Houze (2012) and Kirshbaum et al. (2018) for reviews on the topic. Large low- to mid-level relative humidity, dry air entrainment, and strong as well as deep ascent have been shown to be among the most crucial factors for CI (e.g., Nelson et al., 2022; Marquis et al., 2023). Whether near-ground convergence and ascent occur upstream, over, or downstream of a mountain ridge (Fig. 3) broadly depends on the speed of the flow impinging on the terrain obstacle, the stability of the atmosphere, and the shape and dimensions of the ridge (e.g., Kirshbaum et al., 2018; Imamovic et al., 2019). However, in reality, individual flow and moisture perturbations are extremely complex and subject to strong diurnal cycles (e.g., Zardi and Whiteman, 2013; Adler et al., 2016). On this local scale, terrain-induced flows (e.g., Schmidli, 2013; Panosetti et al., 2016), land use and urban heat islands (e.g., Niyogi et al., 2006; Zhao and Wu, 2018; Liu and Niyogi, 2019), as well as outflow from existing storms strongly influence the convergence regions in addition to the larger-scale flow. Furthermore, mountain slope has a strong influence on the advection of moisture and thereby on CI likelihood (Göbel et al., 2023).

The reliable role of these processes in CI has been shown to increase predictability of convective precipitation over mountains compared to flat terrain (e.g., Bachmann et al., 2020; Khodayar et al., 2021). However, numerical models still often struggle to resolve these processes despite continued improvements in recent decades. Since small details in CI timing and location can have a profound impact on the evolution and organization of severe convective storms, they remain among the main causes for forecast errors. For that reason, CI is a very important research topic to many of the institutions who contributed to this article and a focus of current and recent research (e.g., the past COPS and RELAMPAGO campaigns and the current TEAMx campaign; Wulfmeyer et al., 2011; Nesbitt et al., 2021; Serafin et al., 2020; Rotach et al., 2022).

Severe convective storms typically occur on days with relatively large CAPE, some CIN, and a strong background flow. Within TIM, the CI mechanisms explained above could be narrowed down further when concentrating on these scenarios (e.g., Nelson et al., 2022). This includes a better understanding of CI location relative to mountains (e.g., Nisi et al., 2018; Manzato et al., 2022b) by documenting the boundary layer evolution, thermal circulations, and convergence lines before and during CI. The interplay of dry lines (De Martin et al., 2023), cold fronts (e.g., Pacey et al., 2023), triple points, sea-breeze circulations, frontal waves, and gravity waves (e.g., Mulholland et al., 2020) on CI and storm intensity with complex terrain requires further research as well. For instance, the pronounced maximum in large hail reports over Northern Italy (Fig. 1) and the occurrence of tornadoes in the same region (Bagaglini et al., 2021; De Martin et al., 2023) has been proposed to be partly due to the combined effects of mesoscale boundaries and sea-breezes separating the dry mountain boundary layer from the marine boundary layer over the warm Adriatic and Ligurian Seas.

Another important scientific question concerns scenarios in which CI is suppressed. It has been noticed on several days with high severe convection potential in central Europe that thunderstorm development is forecast but fails or happens later in the day. Our hypothesis is that in these scenarios weak synoptic forcing is supplemented with high concentrations of Saharan dust, which is often advected together with high-CAPE airmasses (e.g., Seifert et al., 2023). The dust decreases radiative warming at the surface and increases warming aloft (Stanelle et al., 2010), which may reduce CAPE. These processes are not implemented





in most operational weather models (e.g., Seifert et al., 2023). Assimilating existing aerosol measurements (e.g., Eirund et al., 2022) could reduce forecast errors of severe convection.

More generally on the topic of aerosols, their role in cloud microphysics and dynamics is a focus of current atmospheric science research from a variety of angles (e.g., Morrison et al., 2020). Regarding severe convection, aerosol concentrations can have a direct influence by acting as cloud-condensation or ice nuclei (CCNs or INPs), thereby influencing hydrometeor
concentrations and thus storm dynamics (e.g., Allen et al., 2020; Varble et al., 2023; Barthlott et al., 2024). Based on this mechanism, cloud seeding for hail suppression is still widely used in Europe, despite scientific evidence pointing against its effectiveness (e.g., Browning and Foote, 1976; Knight et al., 1979; Auf der Maur and Germann, 2021). Also disproved but still used are hail cannons (Wieringa and Holleman, 2006). Given the dramatic increase of hail damage in recent years, more effective ways of damage prevention should be pursued and funds re-allocated. A better understanding of aerosol effects, as
well as outreach and education as part of TIM, could help in that regard.

### 2.4 Storm-scale interaction with complex terrain

How the internal structure of thunderstorms changes in reaction to terrain shape is not well understood due to the lack of high-resolution observations within or around thunderstorms. Some insights exist from idealized modeling studies. Based on qualitative experience with storms in the southern pre-Alps, Feldmann et al. (2024) recently analyzed the interaction of a
supercell storm with a simplified ridge and sloped terrain in highly idealized simulations. They showed that channeling of storm-generated outflow and inflow can affect baroclinic horizontal vorticity generation, low-level wind shear, and storm-relative inflow, all of which have an influence on supercell structure and intensity (e.g., Rotunno and Klemp, 1985; Peters et al., 2020). These findings are largely consistent with previous idealized simulations (e.g., Markowski and Dotzek, 2011; Scheffknecht et al., 2017) and the general observation that supercells can traverse and intensify over mountain ranges (e.g.,
Kvak et al., 2023; McKeown et al., 2024). What seems most needed now is to validate the simulated dynamical processes in real supercells over a large spectrum of cases and types of terrain, an endeavor that requires considerable effort and coordination. Similarly interesting are some non-supercell storms, such as pulse storms (Miller and Mote, 2017), which can produce large amounts of hail for a short time for unknown reasons and which are difficult to anticipate.

On larger spatial scales than individual storm cells, several studies have investigated the interaction of mesoscale convective
systems (MCS) with orography. These systems are typically driven by lift along the gust front of the cold pool (Markowski and Richardson, 2010, chapter 9), which can be decelerated, accelerated, or blocked by mountains, thereby leading to phases of updraft and precipitation intensification and weakening, respectively (Frame and Markowski, 2006; Reeves and Lin, 2007; Letkewicz and Parker, 2011; Smith et al., 2014; Pucillo et al., 2019; Wu and Lombardo, 2021; Lombardo and Kumjian, 2022). Hence, mountains play important roles in upscale growth from single cells to MCS formation, the system's movement, and in
some cases also its decay.

The precipitation enhancement of MCSs over orography is one of the reasons for flooding events in mountainous regions (e.g., Saharia et al., 2017). Another reason is that the lift mechanisms near terrain can lead to repeated re-formation, i.e., "backbuilding" of convective cells, which means that precipitation accumulates over the same location over a long time (e.g.,



Soderholm et al., 2014; Panziera et al., 2015; Kirshbaum et al., 2018). The additional lift along gust fronts of storm outflow,
which is influenced by the terrain slope and the environmental temperature and wind profile, plays an important role in these
scenarios (e.g., Miglietta and Rotunno, 2014).

    Even though past research has helped to increase forecast accuracy, high-precipitation events remain difficult to predict and
are in the focus of many weather services in the Alpine and Mediterranean region based on our partner institution survey.
Unfortunately, they are also difficult to observe accurately. For example, the good radar coverage over central Europe is es-
sential in monitoring high-precipitation events. However, radars have weaknesses over complex terrain. Radar beam blocking
(e.g., Kaltenboeck and Steinheimer, 2015) as well as large vertical gradients of precipitation below the melting layer (e.g.,
Chen et al., 2023) often lead to underestimates of the Quantitative Precipitation Estimation (QPE, Ryzhkov et al., 2022). As
discussed further below, high-resolution precipitation surface and vertical profile measurements as well as gap-filling radars,
aimed to be collected in TIM, could be used to improve correction algorithms of QPE and help in flooding nowcasting and
response.

    Similar accelerations of thunderstorm outflow as in MCS can be expected for downbursts impinging on and channeled
through complex terrain. Downbursts are locally descending jets which can cause damaging winds when colliding with the
surface and spreading out horizontally from the high-pressure region of the cold pool, and which give rise to toroidal vortex
structures (e.g., Fujita, 1990; Canepa et al., 2023). Due to their small spatial and temporal scales, the dynamics of downbursts
over complex terrain are not well understood and require further research with high-density observations and simulations (e.g.,
Canepa et al., 2020).

## 2.5   Storm impacts on infrastructure and society

Damages from severe convective storms with tornadoes, wind gusts, hail, flash floods, or lightning, including secondary impacts
such as landslides, can be immense. Developing a better understanding of these impacts is a prerequisite to developing adequate
measures to mitigate losses. A subtopic that has received renewed attention in recent years is the mechanical or engineering
perspective of severe convective hazards. Regarding tornadoes, ESSL has recently proposed an international Fujita (IF) scale
for tornado ratings based on wind damage indicators that can be used internationally, not just in the US where the enhanced
Fujita scale is used based on US infrastructure (ESSL, 2023). More data from tornado damage surveys will continue to improve
the IF scale in the future. In contrast, flood damage in complex terrain is strongly linked to rainfall-runoff processes and hence,
by nature, a hydrological problem to which meteorological output can only provide background data (e.g., Ravazzani et al.,
2016; Marvi, 2020; Merz et al., 2020).

    Hail is the dominating source of SCS damage. The collection of large hailstones, 3D scanning and analysis of their internal
layers, or study of impact kinetic energy on structures such as roof shingles is increasingly pursued by different groups such
as the Insurance Institute for Business and Home Safety (IBHS) in the US (Brown and Giammanco, 2013), the Northern
Hail project in Canada (Brimelow et al., 2023) and the Bureau of Meteorology in Australia (Soderholm and Kumjian, 2023).
However, Schmid et al. (2024) recently showed that the lack of accurate information on hail stone occurrence, size, and
distribution is hampering damage assessments and risk modeling efforts. With sensitive structures such as solar panels and





wind turbines spreading quickly, such studies are especially important to quantify the damage and loss that can be expected from hail, wind, or their particularly damaging combination: wind-driven hail, which is currently still difficult to predict.

The comparison of large datasets of the 3D scanned and dissected hailstones between different continents, elevations, or climatic regions could also provide valuable insights into hail growth and microphysics within thunderstorm clouds and their relationship to the atmospheric background environment (Soderholm and Kumjian, 2023).

Furthermore, the impact of severe weather on society is strongly dependant on human behaviour and risk perception (e.g., Ripberger et al., 2019). Unfortunately, education on severe convection storms is very limited throughout Europe, perhaps

because of the relative rareness of such events and limited cross-border exchange. This may affect people's trust in severe weather forecasts and how they perceive a severe weather threat. For instance, appropriate safety measures and behavior when encountering tornadoes, large hail, strong winds or lightning are typically not common knowledge. ESSL and the TIM partners share the goal to improve these conditions through exchange about better warning practices as well as through education and outreach at TIM measurement sites and in the media during the field campaign. Furthermore, the topic of severe convective

storms is also somewhat underrepresented in university courses, especially compared to the U.S. where most severe weather research groups are situated. TIM aims to engage students from European universities in the field campaign and thereby help to build the next generation of severe weather forecasters and researchers.

## 2.6 Improvements in NWP

Numerical weather prediction (NWP) models are an important basis for forecasts of severe weather. Short-term NWP forecasts,

i.e. for the next hours can be blended with information from nowcasting systems, that extrapolate observations collected in real-time (e.g., Bojinski et al., 2023), or the models can be initialized very frequently, e.g. every hour or less, in so-called rapid update cycles (Benjamin et al., 2004) assimilating a range of observations available at that time. Often an ensemble of multiple simulations is performed whereby the initial state or the evolution of the predictions is subtly perturbed. The uncertainty resulting from small differences in initial conditions, CI or storm interactions can thus be reflected (Durran and Weyn, 2016;

Bachmann et al., 2020; Bojinski et al., 2023).

Despite these advances in severe weather forecasting techniques, notable forecast errors can occur when NWP models are not able to correctly simulate convective initiation or the correct type of convective system, which may result from an inability to reproduce local changes in factors like moisture content or wind shear. For instance, on 19 July 2023 a series of record-breaking hailstorms occurred across North Italy as intense supercells initiated over the southern Alps and moved across the

Plains. On that day, operational NWP models either failed to produce the storms, wholly or partially, or underestimated their intensity (see Fig. 4). In this and other challenging cases (e.g., Mandement and Caumont, 2021), a big unknown was how storms will behave once they move out of the mountains and onto the plains: they may intensify or decay. This transition from mountains to plains seems neither well-understood from a storm-dynamics perspective, nor generally well-predicted by NWP models for a variety of reasons.

It is known that NWP models have shortcomings near mountainous terrain as their horizontal resolution is only sufficient to resolve the topography to a certain extent. The smoothing at the smallest resolved scales affects how well the interaction





**Figure 4.** Composite radar reflectivity (OPERA) and ESWD hail reports over northern Italy on 19 July 2023 (green: hail diameter 2-5 cm; orange: hail diameter 5 - 9 cm; red: hail diameter > 10 cm), and +7 hours forecast reflectivity by two operational NWP models: ICON-D2 and C-LAEF. The supercell producing the extremely large hail between 18:00 and 19:00 UTC was missed by ICON-D2. C-LAEF forecasted convection, but with weaker maximum intensity.



between the surface and the lowest layers can be represented. Furthermore, the parameterizations for unresolved surface-atmosphere exchange processes were mostly developed for flat terrain and much coarser grids which introduce considerable errors across steep topography (Ceppi et al., 2013). Moreover, the neglection or simplified treatment of subscale effects and use of simplified climatological fields also have a large impact in complex terrain. This includes challenges of numerical model "grey-zones", where resolved processes and parameterizations overlap (e.g., Chow et al., 2019; Kramer et al., 2020; Kirshbaum, 2020; Wei and Bai, 2024). Lastly, methods used for convective-scale data assimilation are not as advanced as for global models (Gustafsson et al., 2018). Improving terrain-related processes in models is the subject of research within the TEAMx project (Rotach et al., 2022).

Aided by experience from TEAMx but with a focus on convective storms, addressing the following areas within TIM is likely to lead to progress in NWP performance: (1) the characterization of the pre-convective environment across complex terrain by developing better data assimilation procedures (e.g., Gustafsson et al., 2018; Bachmann et al., 2020), (2) improving parameterizations for surface-air interactions (Ceppi et al., 2013; Goger et al., 2019; Rotach et al., 2022), (3) improving data assimilation of ongoing convection in satellite and radar and assimilating data from novel sources. As discussed further in section 2.8, the new generation of geostationary meteorological satellites (MTG, Meteosat Third Generation) enables more accurate atmospheric moisture determination and a Lightning Imager similar to the Geostationary Operational Environmental Satellite (GOES) Geostationary Lightning Mapper (GLM) over the Americas (Rudlosky et al., 2020) which can be assimilated into NWP models and used for nowcasting. Quantification of the impact of these new instruments on forecast and nowcast (Leinonen et al., 2023) accuracy is of high importance to guide their use in forecast offices and the development of successor instruments in the future.

Furthermore, the exceptionally high spatial and temporal resolution of TIM-data is of particular interest given that hectometre-scale Earth System modelling is realised in the latest generation of (pre)operational regional NWP models with grid spacings at 1 km and below, e.g., Meteoswiss' ICON-D1 and -D05, as well as planned in global models such as the Extremes Digital Twin by the EU Destination Earth programme (ECMWF, 2023). Field campaign data from drones, surface observations, Doppler wind lidars, or other sources could be tested in model comparison and data assimilation studies (e.g., Sgoff et al., 2022; Nomokonova et al., 2023; Demortier et al., 2023). Since the presence of orography is still often neglected in current NWP postprocessing algorithms (e.g., Schwartz and Sobash, 2017), such observations would also allow the identification of physically-based predictors and targeted observations that could locally improve the calibration of ensemble prediction, nowcasting systems, and machine learning techniques for severe weather warnings in regions influenced by mountains (e.g., Dabernig et al., 2017).

Another potential area of NWP improvement is a better handling of microphysical processes within storms (Morrison et al., 2020). Many uncertainties remain regarding the actual concentration, properties and number of particles in the various hydrometeor classes (cloud water, rain, graupel, snow, hail) because of a lack of measurement data as research aircraft tend to avoid the most intense parts of those storms. However, relatively recent innovations including drones, neutrally buoyant sondes, and vertically-pointing radars (Trömel et al., 2017; Sokol et al., 2018) offer new possibilities to collect measurement data that are crucial for the improvement of parameterizations by weather services and the academic community. The importance of these





innovative observations and their potential use in TIM will be explored below. Furthermore, the effects of aerosol loading are relatively poorly understood and their improvement may benefit forecasts (Seifert et al., 2023; Barthlott et al., 2024), especially on days where the aerosol loading is high.

### 2.7 Climatology and Climate Change

Pan-European severe convective storm climatologies have been produced based on different data, such as severe weather reports, lightning, satellite products, and reanalyses (Punge et al., 2017; Taszarek et al., 2019, 2020a; Manzato et al., 2022b; Lombardo and Bitting, 2024). These studies agree relatively well on a main convective season in summer over most of Europe, except over Mediterranean regions, where activity peaks in fall but with relatively low severity (with the exception flood events). As discussed in the introduction, summertime hotspots for severe convective storms are found near mountain ranges, for example around the Alps, the Pyrenees, the Massif Central, the Black Forest, the Dinaric Alps, and the Carpathian Mountains. Details of this severe storm clustering are still not fully understood, also because public reports of severe weather suffer from biases such as spatial heterogeneity. Continued growth of report networks like ESWD for example by supporting public user reports across Europe but also comparison with diverse pan-European datasets such as from satellites, radars, and lightning networks will provide further details on where severe storms typically initiate, become severe, and decay (e.g., Nisi et al., 2018; Manzato et al., 2022b).

New data gathered in TIM is limited in time and cannot directly assess these climatological features, long-term variability, and effects of climate change. Yet, results from a constrained field campaign can be of immense value for these research areas by improving process understanding or probing "analog" storms that occur in conditions presumably representative of future climate conditions (e.g., Lasher-Trapp et al., 2023).

As one effect of global warming, the increase in average temperature throughout the troposphere and associated scaling in moisture content is generally expected to increase CAPE and hence thunderstorm severity (e.g., Romps, 2016; Agard and Emanuel, 2017; Martín et al., 2024). This upward trend is robust for Europe both for past and projected trends of CAPE or related thermodynamics parameters (Mohr and Kunz, 2013; Mohr et al., 2015; Púčik et al., 2017; Rädler et al., 2019; Battaglioli et al., 2023), as well as in severe weather reports. Stronger updrafts in high CAPE environments are one factor increasing the likelihood of flash floods, in addition to higher moisture content and slower storm motion (e.g., Kahraman et al., 2021). In its Sixth Assessment Report the Intergovernmental Panel on Climate Change (IPCC) states that climate models consistently and with high confidence project environmental changes that would likely lead to an increase in the frequency and intensity of severe thunderstorms, also involving tornadoes, hail, and winds (Calvin et al., 2023, Chapter 11). However, there is low confidence in the specifics of these projected increases.

Large uncertainties remain for the trend in hail, tornadoes, and convective winds because other impact factors than thermodynamics exist, which could counter-balance the positive trend and which are not as well understood (e.g., Raupach et al., 2021). For one, convective inhibition and entrainment are likely to increase in a warmer climate which could lead to decreases in initiation frequency and storm coverage (e.g., Trapp and Hoogewind, 2016; Chen et al., 2020; Peters et al., 2024). The area affected by hail is also likely to decrease because of an increase in the melting layer depth, although large hail is less affected





and is expected to become more frequent due to the positive trend in CAPE (e.g., Gensini et al., 2024). Complicating this, Lin and Kumjian (2022) have indicated that increases in CAPE do not necessarily cause storms to produce larger hail, although such extreme values of CAPE might be relatively rare in Europe (e.g., Taszarek et al., 2020b). Furthermore, the frequency and the typical thunderstorm organization could change in response to modifications in the larger-scale circulation and resulting

weather patterns (e.g., Mohr et al., 2020; Ghasemifard et al., 2024). This is important because severe convective wind gusts and extreme precipitations are often associated with MCSs while most large hail and strong tornadoes are produced by supercells in the US (Blair et al., 2011; Smith et al., 2012; Blair et al., 2017). For Europe, this is likely similar but less researched in certain regions (Tuovinen et al., 2015; Wapler, 2017; Feldmann et al., 2023).

It is unclear how accurately climate models capture these circulation changes. Another more general limitation of climate

models for convective storm research is that most of them still depend on convection parameterization, although research with convection-permitting climate simulation is currently ongoing (e.g., Lenderink et al., 2024). The accuracy of climate models across mountainous areas may also be impaired by systematic errors in modeling surface-atmosphere exchange processes across complex terrain (e.g., Rotach et al., 2022). The record-shattering reports in recent years (e.g., Banerjee et al., 2024) give the impression that at least in some regions global warming likely has a strong influence on severe convective storm frequency.

Interestingly, other observations with periods long enough for climate trend analysis, such as from lightning data or hail pads, indicate no clear upward trend thus far (Manzato et al., 2022b, a; Augenstein et al., 2023), although it is somewhat unclear how representative they are for severe storms in general.

In short, more research beyond the trends of convective parameters such as CAPE seems required to gain more robust knowledge on trends in severe weather. This is where even a relatively short-term field campaign can help because a better

fundamental understanding of the storm dynamics, microphysics, CI, and the production mechanisms of severe convective hazards is needed to know what signals to look for in climate projections and to anticipate how these processes may change in a warmer climate.

## 2.8 Innovative observations and their use

Advances in severe storm research and forecasting have always been linked to development and improvements of new ob-

servational techniques such as radar or satellite (e.g., Brooks et al., 2019). Many new methods and instruments are currently being developed or implemented which could likewise improve our understanding of severe convective storms. As discussed further in this section, this includes new polarimetric radar algorithms, drone-based observations, swarmsondes, microwave links, and new instruments on the Meteosat Third Generation (MTG) geostationary and later on the Metop Second Generation (Metop-SG) polar-orbiting satellites. A field campaign on convective storms in the following years would be ideal to test these

new platforms, validate their performance by comparing different observations of the same cases, and gather new insights. By conducting such a field campaign, technologies, and instruments will again be improved and refined in return as well.

For one, the evolution of storm-top features as seen from satellite imagery has been shown to be strongly linked to storm severity. These features include above anvil cirrus-plumes, warm wake areas, overshooting top properties, or storm-top divergence (e.g., Homeyer et al., 2017; Bedka et al., 2018; Murillo and Homeyer, 2019; O'Neill et al., 2021; Murillo and Homeyer,



2022; Scarino et al., 2023). Questions remain how these characteristics compare in imagery from GOES and the MTG satellites and how indicative these features are in rapidly changing conditions near complex terrain. Novel and more precise information on cloud top properties, cloud-ice content, precipitation and aerosol properties will be provided by the METimage, Ice Cloud Imager (ICI) and the Multi-Viewing Multi-Channel Multi-Polarisation Imaging (3MI) instruments on the polar-orbiting Metop-SG satellites as of 2026.

Lightning is another type of dataset that can be detected remotely with a variety of methods and provides valuable information about the microphysics and severity of a thunderstorm. The new satellite-based MTG lightning imager (LI) will require validation with ground-based lightning sensors before it can be used in research or assimilated in forecast models. In addition to the LI and 2D lightning detections, Lightning Mapping Arrays (LMAs) can be deployed to study the 3D electrification and polarity in thunderstorms. Many mechanisms have been proposed (e.g., Pruppacher and Klett, 2010) and it has been shown that

even slight differences in the environment can have profound effects on cloud electrification, which is an active field of research (Carey et al., 2003; Carey and Buffalo, 2007; Fuchs et al., 2015; Schultz et al., 2015; Chmielewski et al., 2018; Ringhausen et al., 2024). Examples to be investigated in TIM are changes in the lightning activity, polarity or flash size (e.g., Figueras i Ventura et al., 2019; Erdmann and Poelman, 2024) of the storms that traverse complex terrain and undergo rapid changes in response to environmental characteristics such as CAPE, vertical wind shear, cloud base height, or warm cloud depth. These

could be linked to microphysics of the storm assessed through polarimetric radar (e.g., Kumjian and Ryzhkov, 2008), hail collection, and in-situ observations within clouds. A better understanding of these processes would allow for the development of parameterizations and lightning prediction in NWP models (e.g., Cummings et al., 2024).

Furthermore, vertical profiles of the atmosphere are invaluable for severe storm forecasting and research, because they allow for quantification of the ingredients for severe storm (e.g., Doswell, 1987; Doswell et al., 1996). As touched on in sections 2.1

and 2.3, the distribution of moisture both vertically and horizontally plays an important role for convective storms. The new 0.9 micron channel of the MTG Flexible Combined Imager (FCI) instrument for the first time allows for the quasi-continuous monitoring of total column moisture content including in the lowest few hundred metres of the troposphere, with sampling rates over Europe of 10 minutes (available now) down to 2.5 minutes (by around 2027). Such data could help forecast new cell formation and thereby flooding by back-building convection as well as local boundaries which are known to influence CI and

storm severity. Hence, such horizontally homogeneous moisture data is highly desirable, but verification of the MTG retrieval algorithms with in-situ data at high spatio-temporal resolution is necessary, especially near complex terrain. Similarly, the new geostationary infrared sounder (IRS) on board of the MTG-S1, which can also provide vertical profiles of temperature and moisture, requires comparison with high-frequency classical in-situ soundings and ground-based radiometer soundings near orography. The potential value of IRS as a complement to standard soundings has been evaluated at the ESSL Testbed in the

quasi-operational setting (Groenemeijer, 2019). Although there are limitations to detect low-level moisture near the ground with this data, availability of infrared soundings from geostationary orbit at 30-min resolution over Europe will add significant value and overcome the current limitations on temporal resolution of such data from polar-orbiting instruments such as IASI on Metop.





Recent alternative profiling methods are commercial microwave links (CMLs) and uncrewed aircraft systems (UAS or
*meteodrones*) As an opportunistic remote-sensing tool, CMLs typically used in cell towers are rapidly evolving and E-Band
CMLs are increasingly installed in several countries to enable 5G. E-band CMLs can provide information on water vapor
variability (e.g., Fencl et al., 2021). Research is needed to fully exploit the fast-developing K-, E-, and W-band commercial
microwave link (CML) networks for use in precipitation and water vapor monitoring in terms of retrievals and direct data
assimilation at different frequencies. Regarding UAS, in-situ aerial measurements with Meteodrones of the lower troposphere
were tested as high up as 6 km above ground (Guay et al., 2023), compared against radiosonde and remote-sensing instruments
(Hervo et al., 2023) and found to meet different WMO requirements (WMO, 2024). Meteodrones are of interest for vertical
profiles of temperature, humidity, wind speed and wind direction and can operate in horizontal wind speeds of up to 25 m s$^{-1}$.
Despite the limitations of drones due to strict airspace regulations, they could offer a low-cost way to collect high-resolution
3D data that are difficult to attain with conventional point measurements.

Other drone applications are photographic documentation purposes. Cloud, dust, rain- and hail-shaft development can be
observed in time-lapse mode without the limitations imposed by buildings or vegetation being obstacles to the ground-based
observers. Photographic post-event analysis of wind damage, rain damage, and hail damage as well as analysis of hail size
distribution are feasible based on images taken from drones (Soderholm et al., 2020; Lainer et al., 2024).The importance of
aerial imagery including aeroplanes and drones was also stressed in damage survey guidance (Holzer et al., 2023). High-
resolution satellite imagery can be used in a similar way for some applications (Shikhov and Chernokulsky, 2018; Gallo et al.,
2019; Kunkel et al., 2023).

The global advent of dual-polarization radar data in recent decades has enabled the identification of specific polarimetric
signatures linked to the dynamic and microphysical processes at the heart of severe thunderstorms (e.g., Kumjian and Ryzhkov,
2008; Kumjian, 2013; Dawson et al., 2014). Thanks to these observations, novel algorithms have been developed to enhance
the estimation of storm severity, like the automated detection and appropriate assimilation of columns of enhanced differential
reflectivity ($Z_{DR}$) (Snyder et al., 2015; Schmidt, 2020; Trömel et al., 2023). However, radar data quality is often limited near
complex terrain (Feldmann et al., 2021; Germann et al., 2022). For many radar diagnostics related to storm severity like
Probability of Hail (POH, Delobbe and Holleman, 2006), Vertical Integrated Liquid (VIL, Greene and Clark, 1972), Maximum
Estimated Size of Hail (MESH, Witt et al., 1998), or $Z_{DR}$ columns, a high vertical resolution is needed, and beam shielding
by mountains is problematic. Another challenge is the lack of low-level data. A lot of the radar storm signatures, such as hook
echoes or the $Z_{DR}$ inflow arc (Fujita, 1958; Kumjian and Ryzhkov, 2009), are at very low levels and generally not observed in
mountains. Similarly, QPE in mountains can lead to errors in flood assessments due to strong reflectivity gradients below the
melting level (e.g., Chen et al., 2023). In a field campaign, the deployment of mobile radars could target the observational gaps
that currently exist to quantify errors in operational networks. Such an approach was tested in France and Switzerland and can
systematically deliver highly resolved thunderstorm data (e.g., Grazioli et al., 2019). Similar gap-filling data gathered in TIM
can open pathways for improved estimation of storm severity diagnostics. This includes the polarimetric variables introduced
above but also 2D and 3D wind fields within storms, which are known to have a strong influence on hail and tornado potential
(Dennis and Kumjian, 2017; Sessa and Trapp, 2020, e.g., ).





Lastly, *swarmsondes* (Markowski et al., 2018) are balloon-borne probes that act as a pseudo-Lagrangian drifter and that
are drawn through the storm. Swarmsondes can be released in dozens or potentially even hundreds per thunderstorm and
were found to fill the lack of reliable, aboveground, thermodynamic observations near and in convective storms. Swarmsondes
in Europe were first used in the Swabian MOSES campaign in 2021 (Kunz et al., 2022). A slightly adapted version called
*Hailsonde* is currently being used as part of the LIFT campaign in Europe and the Northern Hail Project in Canada to mimic
and analyze the trajectory and growth of falling hailstones within thunderstorm updrafts (Brimelow et al., 2023; Kunz et al.,
2024; Soderholm, 2024). Lack of such data from within thunderstorms is perhaps one of the biggest limitations in the field
of convective storms. Much of our knowledge and model parameterizations of microphysical properties, such as hydrometeor
concentrations within storm clouds, is based on storm-penetrating flights of the T-28 aircraft, which was retired around 20
years ago (e.g., Allen et al., 2020). Hence, gathering such data with aircrafts or sondes in a field campaign could allow for big
leaps in our understanding and forecast capability.

## 3   Synergy and rationale for the TIM campaign

As reviewed in section 2, a wide variety of research topics (RT) exists regarding severe convective storms. Fig. 5 summarizes
these subjects and highlights their synergies. The importance of this research is emphasized by the observed increase in severe
storm losses in recent years as well as the projected increase in severe storm activity because of climate change. ESSL is
therefore coordinating the TIM field campaign as a first-of-its-kind Pan-European campaign on severe convective storms with
a focus on mountain-related processes. To be highlighted as main research questions for the TIM campaign are

(1) Why do severe weather reports cluster near mountain ranges? To answer this, the terrain-induced perturbations of
environmental parameters like CAPE and wind shear will need to be weighed against storm-scale processes and CI in
different regions and scenarios. (RT 1,2,3,4)

(2) What are gaps in our understanding of the storm-internal processes leading to large hail, strong winds, lightning,
tornadoes, and extreme precipitation and how are these processes modulated by the orography? Such basic understanding
is also essential to better anticipate the impact of global warming on severe storms. (RT 1,4,5,7,8)

(3) What are the strengths and limits of new observation techniques from satellites, radars, or drones? TIM will provide
the unique opportunity to compare these datasets to high-resolution sampling of individual storms and of heterogeneous
environments near complex terrain. (RT 2,4,6,8)

(4) What observations are most important to improve numerical weather prediction models and nowcasting? High-
resolution datasets of the pre-storm environment and thunderstorm properties in TIM can be used in model studies to
help answer this question and also to improve parameterizations of orographic or microphysical processes. (RT 1,4,6)

Detailed planning of the campaign and specific strategies targeting these and other research questions will continue over
the next years. Here, we only provide a first impression of the general campaign concept. Table 1 and Figure 6 provide an



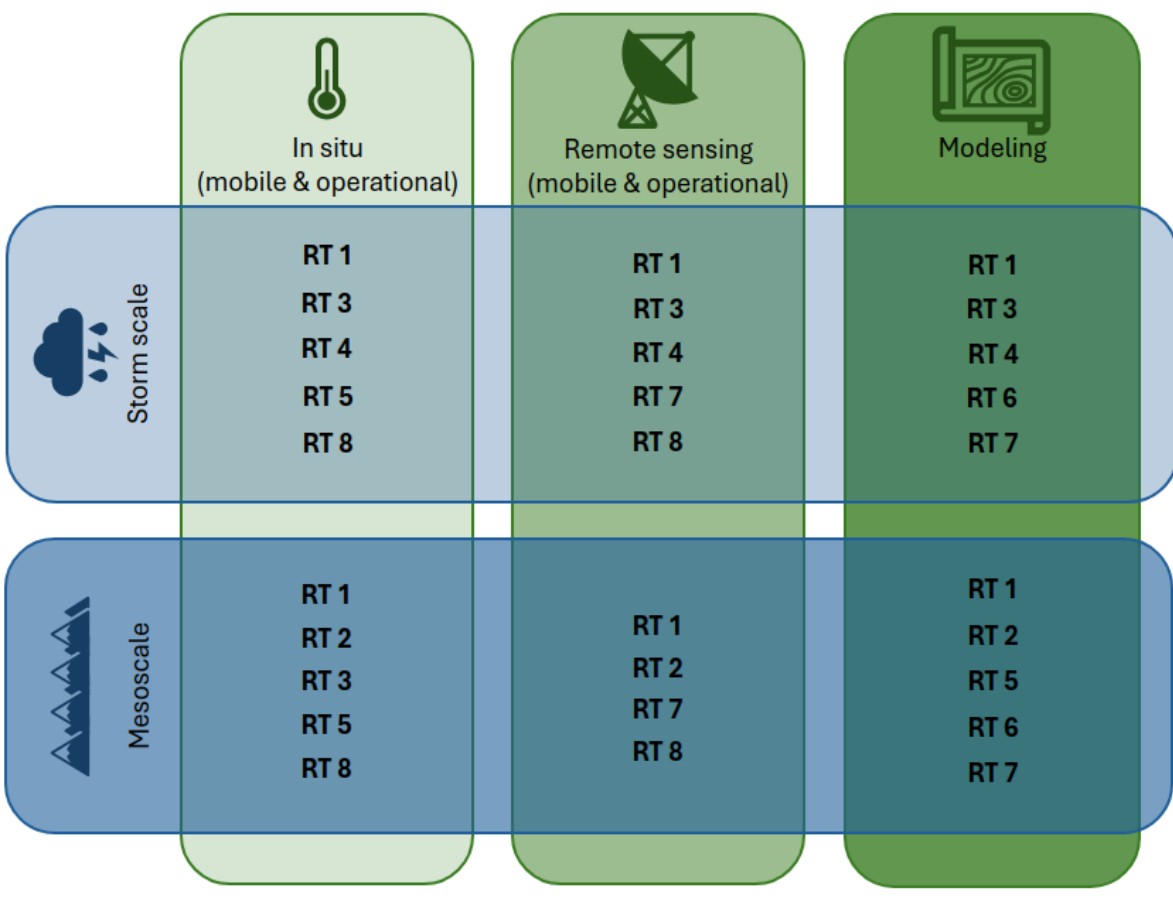

**Figure 5.** Overview of the research topics in TIM and their synergies across methods and scales

exemplary overview of the planned data collection. The TIM campaign can profit from the wealth of already operational obser-vations across Europe (red text in Fig. 6). This includes meteorological stations with ground data, regular vertical soundings, radars, satellites, lightning sensors, and public reports. Some of these datasets are currently not openly available to researchers. One example is the European radar network, which is globally unique in its density and coverage and provides excellent opportunities for analysis over a large region but is often only analyzed for each country separately. Data sharing in TIM could therefore accelerate the development of new radar-based algorithms for severe weather prediction, for instance based on machine learning (Leinonen et al., 2023; Forcadell et al., 2024). The open-data approach of TIM provides a pathway to make datasets available to participating researchers and to include early feedback into what data structures are most useful,



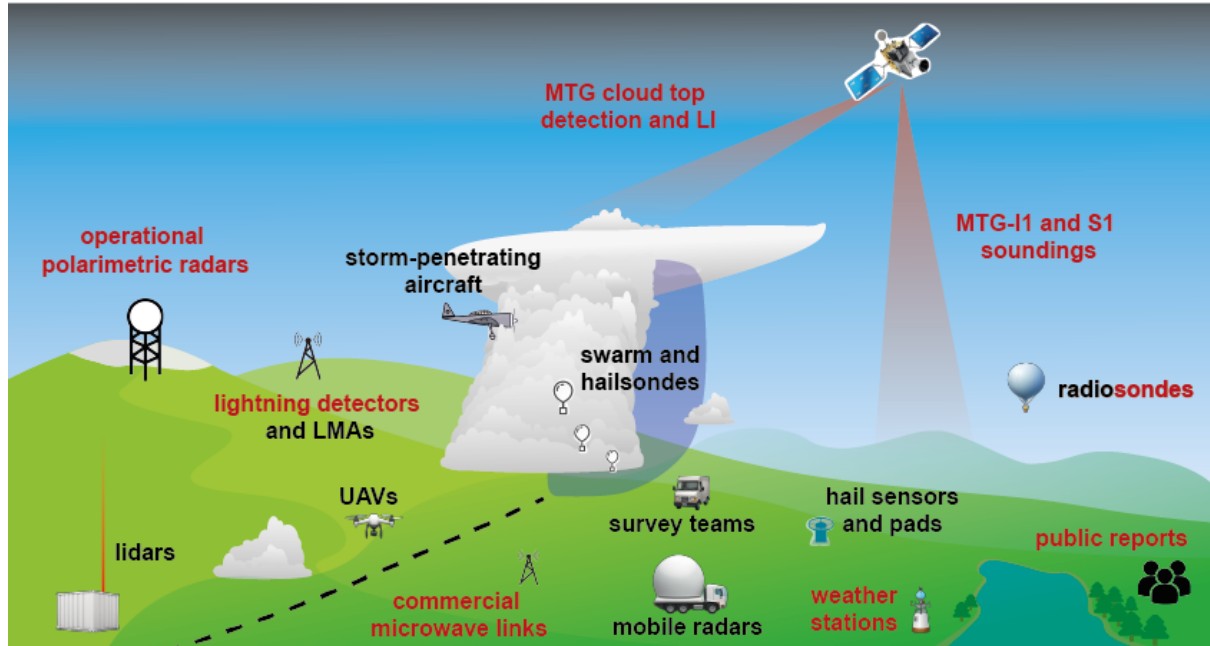

**Figure 6.** Overview of the observations planned in TIM. Since the campaign is still in an early phase, this can be seen as an exemplary setup. Red colored text denotes already or soon-to-be operational data while black text would be part of TIM operations.

for example for machine learning scientists. Ideally, TIM will be a first step towards permanent sharing of such datasets in the future.

However, additional data from the TIM campaign is needed, especially in and around thunderstorms and at higher spatial and temporal resolution than currently provided by operational networks (black text in Fig. 6). This includes mobile radars, swarmsondes, drones, high-resolution soundings, and survey teams. Such observations are often difficult due to the fast-evolving and small-scale nature of convective storms. Thus, to sample a high number of cases it is necessary to be mobile. On the other hand, some of the research objectives described herein require specific measurements which can only be supplied by fixed instru-

ments, such as lidars, hail sensors, LMAs, or high-density station networks, which cannot be re-located on a daily basis (e.g., Emeis et al., 2018). A field campaign with both mobile and fixed-site measurement strategies offers the unique opportunity to address this dilemma.

To sample individual storms and their hazards, TIM will deploy several mobile teams that can travel to a region where highimpact storms are expected on a given day. At the same time, multiple TIM focus sites will be set up over the convective season

to continuously monitor the same region, supplemented by mobile teams when severe storm activity is expected in the same location. A similar strategy has been successfully used in several campaigns focused on convective storms in target regions (e.g., PERILS, RELAMPAGO). A unique aspect of TIM is that focus sites will be maintained by participating institutions based on individual funding and therefore tailored to the specific research interest in these regions. In parallel, core funding



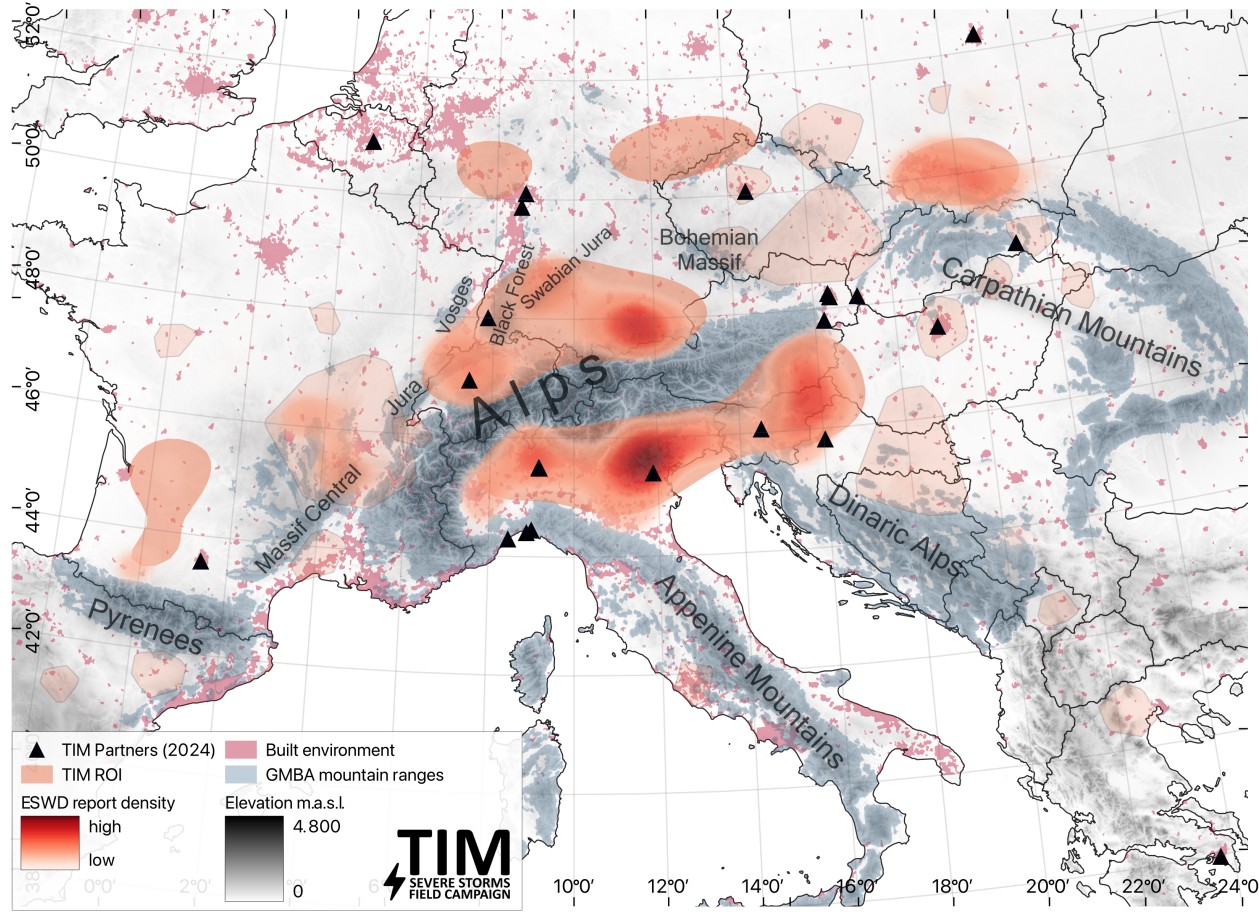

**Figure 7.** Map of the TIM domain with regions of interest (ROI). ROI were identified based on in-domain ESWD severe hail reports (c.f. Fig. 1). Main ROI are primary hotspots of report density, while secondary ROI highlight further distinct clusters of reports (see text). Also shown are mountain ranges of the GMBA mountain inventory (Snethlage et al., 2022b, a) and built-up surfaces (Pesaresi and Politis, 2023). Black triangles show official TIM partner institutions headquarter locations as of May 2024.

of TIM will support mobile teams to collect additional high-resolution where they have the highest research use. This flexible

approach allows the effective use of funds and overlap of different research topics (Fig. 5).

Figure 7 highlights potential regions of interest (ROI), where ROI are identified through clustering analyses of ESWD severe hail reports in the TIM domain. While storm reports can be subject to reporting biases, ESWD report densities show good agreement with existing climatologies and using the large ESWD database allows to identify regions of high activity where interest and potential impacts are most relevant. Primary hotspots with very high report densities delineated by kernel-density

estimation (KDE) are apparent along the central northern and central-to-eastern flanks of the European Alps including the Jura and Swabian Jura as well as to the North of the Carpathian Mountains. Relatively fewer but yet distinct clusters grouped by



density-based spatial clustering of applications with noise (DBSCAN) become apparent in the vicinity of medium mountain ranges such as the Massif Central, the Bohemian Massif and the Dinaric Alps. Which focus sites can ultimately be investigated in depth with TIM will depend on the priorities and funding of individual partner institutions. Besides geographical and cli-
matological differences among the regions, also the densities, lengths and types of operational observations vary substantially. TIM aims at addressing these regional differences both in the planning and interpretation of the results.

The collaboration between many institutes brings together the necessary expertise to handle these diverse data needs, which no single institution can cover on its own. TIM already has a long list of partner institutions officially engaged who also contributed to this article (see next section and Fig. 7). ESSL is leading these efforts to ensure that possible conflicting interests
over the wide spectrum of institutions can be coordinated. Because of this spectrum of interests and because of the high spatio-temporal variability of convective storms (e.g., Taszarek et al., 2020a) the campaign is planned for three consecutive convective storm seasons, currently from 2026 to 2028. This is followed by two years of data analysis to ensure the effective and coordinated use of the observations. This observational data will be supplemented by numerical modeling experiments tailored to the respective research questions and focus regions. Furthermore, ESSL testbeds during and after the campaign
provide ample opportunities to foster exchange with European severe weather forecasters.

High-quality forecasts focused on severe convective storms are also essential for the coordination of the campaign. Like in previous severe storm campaigns, a forecast team will be responsible for briefing the research groups on an active campaign day. The forecasts can benefit from the bundled resources of partner institutions and experience of projects like the European Storm Forecast Experiment (ESTOFEX), Previsione Temporali (PRETEMP), as well as local forecast offices who know the
characteristic meteorology of a certain mountain range.

Thanks to the globally-connected severe storms community, TIM can also profit from the experience of international researchers. Exchange at the recent European hail workshop has already opened fruitful discussions and opportunities for international collaboration. Specifically, TIM researchers could coordinate with, and learn from the planned ICECHIP and TEAMx campaigns (Adams-Selin et al., 2022; Rotach et al., 2022). Their research foci (hailstorms in the US Great Plains and terrain
effects on the atmosphere) could be complemented and linked by TIM's focus on severe convection near mountains. Further coordination with the Northern Hail and Tornado Projects (e.g., Brimelow et al., 2023), the LIFT campaign (Kunz et al., 2024), and observational efforts of the Australian Bureau of Meteorology (e.g., Soderholm, 2024) are planned. TIM has the ambition to foster international collaboration, educate through public outreach, and introduce students to the field of convective storms, which will ultimately help to meet the challenges that the increasing risk of these phenomena pose to European society.



**Table 1.** Preliminary list of potential TIM observations during the active field campaign. The list is non-exclusive and not necessarily to be realized at all locations.

| Data | Details |
| --- | --- |
| Radiosondes or Meteodrones | Vertical profiles of temperature, moisture, and wind with a temporal resolution <1 hour and comparing different terrain-relative locations |
| Satellite-based soundings | Test temperature and moisture profiling and detection of horizontal heterogeneities against in-situ measurements |
| Operational (polarimetric) radars | Nowcasting and analysis of storms, including internal microphysical and dynamical processes |
| Mobile Doppler radars | Higher-resolution data for the above for in-depth dynamics or gap filling analyses |
| Drone surveys | Detection of hail spectra, hail swaths, and damage surveys from hail, tornadoes, and straight-line winds |
| Manual survey temas | Hail collection and manual damage surveys. Launching of Swarmsondes. |
| Public reports | Collection of widespread severe weather reports as ground truth |
| Hail pads or sensors | Same as above for hail with higher accuracy and detailed spectrum information |
| Swarmsondes | Gathering within-storm data such as updraft and downdraft velocity, moisture content, internal boundaries, and hail trajectories |
| Satellite cloud top detection | Analysis of cloud top features and their link to severe weather |
| Satellite lightning mapper | Same as above for lightning |
| Lightning sensor network | See above |
| Lightning Mapping Arrays | Analysis of 3D lightning activity and polarization regions |
| Airborne Doppler lidar and radar | Measurements of 3D winds |
| Storm-penetrating aircraft | Gathering within-storm data such as water content, CCN and INP concentrations |
| Ground-based and airborne lidars | High-resolution vertical profiles of wind shear and boundary layer processes |
| Meteorological station network | High-resolution data near the surface |
| Isotopic water vapor and aerosol collectors | Source analysis of water vapor composition and aerosols |



**TIM Partners**

**Extended co-authors of this article (in alphabetical order):**

Martin Adamovsky[10], Clotilde Augros[12], Ulrich Blahak[13], Vojtěch Bližňák[14], Stephan Bojinski[15], Tobias Bölle[16], François Bouttier[12], Massimiliano Burlando[17], Federico Canepa[17], Orietta Cazzuli[18], Xavier Calbet[19], Alessandro Ceppi[33], Fleur Couvreux[12], Kálmán Csirmaz[21], Tamás Csonka[21], Stavros Dafis[22], Mária Derková[23], Francesco Domenichini[24], Grzegorz Duniec[26], Raquel Evaristo[27], Tomáš Fedor[25], Massimo Enrico Ferrario[24], Michael Frech[12], Enrico Gambini[33], Jaroslav Hofierka[25], Ákos Horváth[21], Adam Jaczewski[26], Kornél Komjáti[21], Vinzent Klaus[28], Zsófia Kocsis[21], Michael Kunz[3], Máté Kurcsics[21], Robert Kvak[14], Konstantinos Lagouvardos[22], Katharina Lengfeld[12], Marco Mancini[33], Marc Mandement[12], Olivia Martius[6], Massimo Milelli[20], Samuel Monhart[31], Gian Paolo Minardi[18], Antonio Parodi[20], Giovanni Ravazzani[33], Didier Ricard[12], David Rýva[10], Tobias Scharbach[27], Stefano Serafin[32], André Simon[23], Miroslav Šinger[23], Zbyněk Sokol[14], Gabriella Szépszó[21], Silke Trömel[27], Benoît Vié[12], Clemens Wastl[29], Christoph Wittmann[29], Petr Zacharov[14], Matteo Zanetti[18], Dino Zardi[30]

**Official Partner institutions of the TIM campaign who have signed letters of intent (as of 22/08/2024):**

Albert-Ludwigs-Universität Freiburg (University of Freiburg)

[29]Austrian Federal Institute for Geology, Geophysics, Climatology, and Meteorology (GeoSphereAustria, former ZAMG)

Austro Control (AustroControl)

[20]CIMA Research Foundation (CIMA)

[14]Czech Hydrometeorological Institute (ČHMÚ)

[13]Deutscher Wetterdienst (DWD)

Državni hidrometeorološki zavod (DHMZ)

[15]European Organisation for the Exploitation of Meteorological Satellites (EUMETSAT)

[21]HungaroMet Hungarian Meteorological Service (HungaroMet)

[26]Institute of Meteorology and Water Management - National Research Institute Poland (IMGW)

KU Leuven, Department of Civil Engineering

[12]CNRM, Université de Toulouse, Météo-France, CNRS, Toulouse, France

[22]National Observatory of Athens - Institute for Environmental Research and Sustainable Development (NOA)

[25]Pavol Jozef Šafárik University (UPJS)

[33]Department of Civil and Environmental Engineering (D.I.C.A.), Politecnico di Milano, Milano, Italy

[24]Regional Agency for Environmental Protection and Prevention of Liguria (ARPA Liguria)

[18]Regional Agency for Environmental Protection of Lombardia (ARPA Lombardia)

[24]Regional Agency for Environmental Protection and Prevention of the Veneto (ARPA Veneto) (ARPA Veneto)

[23]Slovak Hydrometeorological Institute (SHMÚ)



Slovenian Environment Agency (ARSO)

[6]University of Bern - Institute of Geography

[17]University of Genoa

**Other contributing institutions to this article:**

[14] Institute of Atmospheric Physics - Czech Academy of Sciences (CAS)

[16]Deutsches Zentrum für Luft- und Raumfahrt (DLR)

[19]Agencia Estatal de Meteorología (AEMET)

[27]University of Bonn

[28]Universität für Bodenkultur (BOKU)

[30]University of Trento, Department of Civil Environmental and Mechanical Engineering (DICAM)

[31]Federal Office of Meteorology and Climatology MeteoSwiss

[32]University of Vienna, Department of Meteorology and Geophysics

*Data availability.* Severe weather reports are available from the European Severe Weather Database (ESWD, eswd.eu).

*Author contributions.* Author Fischer oversaw and organized the writing and provided Figs. 2, 3, and 6. Authors Groenemeijer, Holzer, Feldmann, Schröer, Battaglioli, Schielicke, Púčik, Gatzen and Antonescu equally contributed through writing tasks or Figures. The extended co-authors (see list above) provided initial writing topics including short paragraphs which were incorporated into the article by the first author. They also proofread the manuscript draft and provided references and suggestions on specific topics.

*Competing interests.* No competing interests are present.

*Acknowledgements.* ESSL preparatory work for TIM is supported by the Provincial Government of Lower Austria Science Fund grant number K3-F-190/013-2023. Feedback on an early version of the manuscript by Rebecca Adams-Selin, Karen Kosiba, Matthew Kumjian, Kelly Lombardo, Paul Markowski, James Marquis, Stephen Nesbitt, Bruno Ribeiro, Joshua Soderholm, Jeff Trapp, and Joshua Wurman is greatly appreciated.



**Table 2. Glossary of abbreviations that are repeatedly used**

| | |
|---|---|
| CAPE | Convective Available Potential Energy |
| CCN | Cloud Condensation Nuclei |
| CI | Convection Initiation |
| CIN | Convective Inhibition |
| CML | Commercial Microwave Link |
| ESSL | European Severe Storms Laboratory |
| IF | International Fujita (Scale) |
| INP | Ice Nucleating Particle |
| LMA | Lightning Mapping Array |
| MCS | Mesoscale Convective System |
| MESH | Maximum Estimated Size of Hail |
| POH | Probability of Hail |
| QPE | Quantitative Precipitation Estimation |
| ROI | Region of Intererest |
| RT | Research Topic |
| SCS | Severe Convective Storms |
| VIL | Vertical Integrated Liquid |

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
