# Peer review of "Invited perspectives: Thunderstorm Intensification from Mountains to Plains"

_EGUsphere, 2024_

## Referee Comment (RC1)

EGUsphere-2024-2798

Invited perspectives: Thunderstorm Intensification from Mountains to Plains.

Recommendation: Accept pending major revisions

This article is a discussion about a potential field campaign designed to examine severe weather in the mountainous regions within Europe. Thorough motivation is provided and a real need for the campaign is evident. Many different topics are identified as potential foci of the campaign and of interest to the participating European partner institutions. In fact, essentially every potential severe weather hazard is included, with the only real discriminator the need for the hazard to occur near a mountain. I don't envy the authors their job of needing to synthesize dozens of competing interests into a single article. Unfortunately, that is exactly what the lead investigators of this campaign will need to do to make it successful. The number of research topics included is too large to be addressed by just one campaign, unless the budget is much larger than I'm used to for campaigns (if so, great!). But by trying to address everything, you'll have a big chance of not addressing anything.

I understand that the funding decisions are still very much unknown, and the sources of the funding will dictate (at least in part?) what research topics will be highlighted. However, I highly recommend the following:

1. A lead investigative team needs to be established. It might need to be expanded as additional funding sources are added, but there should be a core team from the start.

2. A process needs to be established to identify what research topics will be focused on. Will the lead team decide unilaterally? Will each contributing institution get to make their own decisions? Perhaps there will be a rotation, with focus on, say, hail storms during one period and severe winds the next period.

3. It reads like the campaign will be a mix of extended instrumentation deployment coupled with more targeted observing periods. How will these more targeted period be decided upon? Who will be in charge of each one? (While obviously they will want input from everyone, this could easily turn into "too many cooks spoil the soup".)

4. How will data sharing be handled? Will one organization host everything, or will there just be agreements that the data will be available (and to whom)? What guidelines will be provided for how available the data must be? Who will decide those guidelines?

The article needs describe how points 1-4 above will be addressed. If they can't be addressed completely yet (e.g., #3 might depend on who participates), explain how it will be addressed and when. I have a few more targeted comments about this same subject.

- Lines 220- 251 (Section 2.4): This section describes at least 3 different connective modes (supercell, non-supercell pulse storms, MCSs) and 3 hazards (hail, flooding, and downbursts). Each of these represent full lines of research in their own right. It will be difficult to avoid each group trying to optimize observations for their interests at the expense of others. As just one example, observational strategies for the 3 hazards listed above are quite different. Hail needs radar scanning maximized in mid-levels with a focus on high resolution dual pol data. Downbursts need it maximized in low-levels; with the range brought in so velocities aren't aliased.

- Lines 331-339: The wide-ranging research problems described in this short paragraph (including aerosol impacts on convection! A whole line of research mired in controversy over whether it even exists!) cannot be solved with the 3 new observation types listed. Drones cannot provide in situ microphysical observations within a storm (unless

European laws are significantly different than US ones, which of course could be a possibility.) Vertically pointing radars, while helpful, require the thunderstorm to traverse directly overhead, so the sample size will unfortunately be small. While I don't doubt that a targeted set of remote sensing and in situ observations, using existing observation technologies, coupled with carefully structured planned DA and modeling experiments could provide improvement in NWP convective microphysical processes, nothing like that is suggested/described herein. It would need its own dedicated campaign/research effort.

- Figure 5: It seems like almost all of the research topics could fit under both scales and all methods. Can you offer some explanation about why topics are placed under one method/scale and not another? I understand you are still working out the priorities of the project, but even identifying topics that are *not* to be included would be helpful here. (E.g., data assimilation is mentioned back in Section 2, but doesn't show up Section 3 – is it being saved for later?)

In addition, I have the following major comments:

- Lines 19 - 27: I would not necessarily conclude from Fig. 1 that the largest hail is concentrated near peaks in terrain. Obviously, the southern Alps is a hot spot, but beyond that reports seem fairly scattered. I don't doubt what you say is the case, but I'd recommend including plots of other datasets, like those you mention on lines 23-24, to support your claim, particularly given your main science question in Line 28. Population biases also need to be accounted for. Further, unless you also want to include global hail data in your intro, I'd change that science question to "why are storms in the vicinity of European mountain ranges more severe"?

- Lines 163- 165: How will the operations plan handle the mobility challenges that happen near orography? What about forecasting challenges?

- Lines 197- 198: The impact of dust (or even CCN) on convection or CI is still uncertain. I would provide more background information to support this specific idea as a hypothesis.

- Lines 200-201: Do we have sufficient aerosol measurements (particularly aloft) to make them worth assimilating?

- Lines 273-282: Great points and ideas! However, I'd like to see more explanation about how they will be carried out as they are hard goals to achieve. Are there specific plans to partner with identified educational groups? What about partnerships with the news media (or other typical disseminations of warning information)? What about the forecasters issuing the warnings themselves, are they comfortable incorporating guidance about recommended safety measures/behavior into their warnings? What additional information might they need to provide more tailored recommendations?

- Lines 340-392: This section has a great description of the many complex problems that can prevent climate models from capturing all the complexities of trends in severe weather. However, one point not explicitly mentioned is the difficulty in translating environmental convective parameters into knowledge of severe weather. Even convection-allowing models that correctly predict the development of thunderstorms still struggle with translating that information into the severe convective hazard itself. I remind the authors that no existing study or method has successfully shown positive skill in forecasting the occurrence of 50 mm hail, for example, and many studies, in fact, have shown no skill at all or even negative skill (e.g., Gange et al. 2017; Adams-Selin et al. 2019, 2023; Gensini et al. 2021). The field campaign plans and increased spatial/temporal observations mentioned in Lines 388-392 may very well uncover new environmental condition - storm dynamics relationships that can improve the situation.

However, it is also highly possible that small, storm-scale processes may be the differentiator between a storm that produces a severe hazard and one that does not; processes that cannot be identified solely by the surrounding environment (e.g., Adams-Selin 2024). If that is the case, it will mean predicting these hazards, and estimating trends in their occurrence solely from relatively coarse environmental fields, may not be possible. I caution the authors to allow for that result.

- Lines 509-512: Yes, I absolutely agree that data sharing will be one of the key outcomes of this effort. It will be so important, in fact, that I'd like to see more detail here explaining my point #4 above.

Minor comments:

- Line 26: What makes these population centers highly vulnerable?

- Line 34: Based on Fig. 2, it seems like the survey pre-assumed the severe storms would be near mountains? Again, not a problem as long as that assertion is better motivated up in the intro.

- Lines 53-57: What about LIFT and Swabian MOSES?

- Lines 90-91: If you included a lightning rate climatology in section 1, being able to reference the spatial differences between it and OTs and/or hail reports would be a good supporting statement here.

- Line 104: These perturbations would be over a deeper layer than just the boundary layer, right? Boundary layer perturbations alone won't produce the modifications you mention.

- Line 109: "could not be"→ "have not been"

- Line 128-129: Re: upslope few on the leeward side. I agree with the statement, but perhaps include a clarifying phrase about how the upslope flow is opposite to the typical prevailing winds (otherwise, of course, it wouldn't be the leeward side.)

- Lines: 162- 163: Expand upon the results in Feldmann et al. (2023) instead of relying upon Fig. 1.

- Line 210: I'd reference the section where outreach and education efforts will be discussed further.

- Lines 221-223: While I agree with your statement here, how frequent are these different types of storms? How difficult will they be to sample, and climatologically, how likely are they?

- Lines 258-259: Are tornado surveys planned as part of TIM?

- Lines 270-272: While comparison of collections of scanned and dissected hailstones between different continents, elevations, etc. are indeed important, they can still only represent a very small percentage of possible hail instances produced by a given storm. Additional observational methods need to be included that will capture larger spatial and temporal areas of the hailswath: e.g., drone survey, time-dependent disdrometer information, a targeted (mobile, if possible) hailpad network.

- Lines 321: Some comparison of the spatial and temporal resolution of currently assimilated observations to the planned TIM observations would be helpful here. Right now the reader doesn't have a good way to assess how impactful these new observations might be. Over how large an area are they planned?

- Lines 326-330: While I agree that more observations are always helpful, my feeling is that determining the location and quantity of such observations to allow for potential improvement of model calibration is not a simple thing. I'd like to see more information from past studies explaining why and how these observations will be configured. Again, one suspects that this goal is worthy of a research project all on its own.

- Lines 407-409: ESA's new EarthCARE satellite, and the upcoming planned NASA INCUS mission, would both be good satellite datasets to incorporate.

- Lines 410-422: The described research topics, particularly validation of LI data via ground-based LMAs, will require deployment of a reasonably sized ground network for a not-insignificant amount of time, particularly if measurements of such precision will be made that can be linked to storm microphysics. Will enough of a network be able to be established that these science questions can reasonably be answered?

- Lines 428, 432: What is the horizontal and vertical resolution of these new profiling instrument datasets? How much of an impact will these resolutions have on the planned science?

- Lines 442-444: Will these new commercial microwave link networks be established near the campaign location(s) of interest? Where do they monitor? What kind of observations are needed for validation/DA studies?

- Lines 448: What type of airspace regulations re: drones are there in the field campaign location(s) of interest? Will the regulations allow measurements to any reasonable depth? (Drones not being able to fly within thunderstorms due to airspace regulations has been a big frustration of mine.)

- Lines 470-474: Do you have plans for how you will prioritize placement of radars? For example, will filling in gaps in coverage, or establishing the best dual-Doppler coverage be more important? How will those priorities be decided?

- Lines 494-495: This sentence encompasses ~5 different lines of research in just the first half of the question!

- Line 500: Observations most important to NWP models and forecasting…. For what purpose? Convection, orography, other?

- Line 501: What is meant by high resolution? (Rough order of magnitude)

- Lines 523, 530: What instrumentation will the mobile teams consist of? How will their focus be determined, both overall and IOP-by-IOP? What types of convection will be prioritized?

- Lines 527-528: Will research topics, in addition to instrumentation, also be based on individual funding?

- Lines 532: What kind of climatologies do the ESWD severe report database agree with?

- Line 549: How will the numerical modeling experiments be designed and coordinated among groups?

- Line 563: Educating through public outreach and introducing students to convective research are great goals, but no details are provided on activities to achieve those goals.

---

## Referee Comment (RC2)

**Review of EGUsphere-2024-2798**

This is an excellent article outlining the arguments for a European field campaign called "Thunderstorm Intensification from Mountains to Plains" (TIM). The paper outlines the high damage potential of severe storms in Europe, summarises the key research topics for TIM to address, and outlines a possible field campaign with focus regions and a synergistic set of instrumentation and research goals. The article is exceptionally well written and compelling. I have only minor comments that should be addressed before the article will be ready to accept. First, references are missing in a few places that I have listed below. Second, while severe storms are affected by climate change and that is a key reason that TIM should go ahead, the language used when writing about the effects of climate change on severe storms needs to be precise about the uncertainties in our current knowledge that exist around certain changes.

**Specific comments**

1. Line 26-27: "are projected to experience the strongest increases in severe weather occurrence as a result of global warming" – I assume this statement relates to projected increases in severe storm environments, which is correct, but since there are possible disconnects between environments and severe weather occurrence (which may also depend on hazard type), care needs to be taken. If the authors mean environments, I would suggest changing this sentence to "increases in atmospheric conditions prone to severe weather occurrence". Also, I think Battaglioli 2023 did not show projection information but rather past trends.

2. Line 107: "but also the terrain shape and dimensions, and the surface use such as cities or cropland" - the authors should include some references to previous work about these factors.

3. Lines 110-113: "Moisture is often relatively low directly over mountain ridges, leading to less CAPE compared to the surrounding slopes and valleys. Local increase in water vapor content can be found along convergence lines, near lakes, or other regions where moist air can be transported or accumulated, which would lead to larger CAPE, all else being equal." These statements require references.

4. Line 120: "Nevertheless, studies have suggested" - it is unclear which studies have suggested these findings.

5. Line 175: "as well as outflow from existing storms" – any previous work on this in particular?

6. Line 178: "The reliable role of these processes in CI has been shown to increase predictability of convective precipitation over mountains compared to flat terrain" – it is unclear to me what is meant here. Is it that the use of variables related to these processes increases predictability?

7. Line 192: "has been proposed" – by who?

8. Line 253: Flash floods should be list as secondary impacts (since they depend on terrain) and the words "flash floods" in this list could be replaced with "extreme rainfall".

9. Line 262: "Hail is the dominating source of SCS damage" – I agree, although the ranking could be regionally dependent and this statement needs a reference.

10. Line 269: The authors should include a reference or two to the numerous wind-driven hail studies that show it is more damaging than non-wind-driven hail.

11. Line 360: "as well as in severe weather reports" – important to include a reference to the study showing this conclusion.

12. Lines 362-364: "In its Sixth Assessment Report the Intergovernmental Panel on Climate Change (IPCC) states that climate models consistently and with high confidence project environmental changes that would likely lead to an increase in the frequency and intensity of severe thunderstorms, also involving tornadoes, hail, and winds (Calvin et al., 2023, Chapter 11)" – while IPCC gives high confidence to CAPE increases in tropics and subtropics, other projections are given with lower confidence (e.g. there is "significant uncertainty" around projected changes in hail, lightning, and tornadoes.) See e.g. Table 11.2. Given the different trends possible for different hazards (e.g. because hail is affected by melting which can lead to decreases in surface hail frequency with increased temperature) and given that increases in CAPE do not necessarily always translate into frequency increases (given offsetting effects), the statement made here by the authors should be rephrased to show the level of uncertainty that exists (I note the next paragraph does a good job of mentioning these factors, but the statement still requires slight revision for accuracy).

13. Line 394: Have advances *always* been linked to new technology? Would "often" be more accurate?

14. Line 464: A reference for $Z_{DR}$ columns for hail detection is required here.

**Technical corrections**

1. Line 87: "quite long distances" is rather vague and an indication of what it means could be given.

2. Line 152: grammatically, "encounter" should be "encountering" here.

3. Line 196: "fails" should be "fails to occur".

4. The authors have used "exemplary" where I think "example" or "possible" is meant ("an exemplary setup" for example means a gold-standard setup, whereas "an example setup" may be what the authors mean).

5. Line 531: a word (data?) is missing here.

---

## Author Response (AR1)

EGUsphere-2024-2798
"Invited perspectives: Thunderstorm Intensification from Mountains to Plains."

**Author comments to Reviewers (in red)**

Summary: We appreciate the suggestions by both reviewers, which made many aspects of the paper clearer. The article structure was not changed but many parts were rephrased. Following the comments of reviewer #1, Fig. 5 was replaced with a mind-map (to provide a clearer overview), and Table 1 was removed (to be less specific about the campaign plans). Figs. 1, and 7 were updated.

**Reviewer #1**

Recommendation: Accept pending major revisions

This article is a discussion about a potential field campaign designed to examine severe weather in the mountainous regions within Europe. Thorough motivation is provided and a real need for the campaign is evident. Many different topics are identified as potential foci of the campaign and of interest to the participating European partner institutions. In fact, essentially every potential severe weather hazard is included, with the only real discriminator the need for the hazard to occur near a mountain. I don't envy the authors their job of needing to synthesize dozens of competing interests into a single article. Unfortunately, that is exactly what the lead investigators of this campaign will need to do to make it successful. The number of research topics included is too large to be addressed by just one campaign, unless the budget is much larger than I'm used to for campaigns (if so, great!). But by trying to address everything, you'll have a big chance of not addressing anything.

We thank the reviewer for their very detailed and helpful comments. Even though the planned budget for TIM would be large, we agree that the full scope of topics presented in this article is too large for one field campaign. The purpose of this article is not to present the final campaign plan or a fully-developed project proposal, which would be impossible at this initial stage and for such a broad range of topics. Rather, the article aims at reviewing the state of knowledge to provide a comprehensive overview over the research topics that need further investigation and *could* be addressed in TIM.

We do not plan to realize all of them but we believe it is important to provide such an overview to emphasize the need for a field campaign and to provide a scientific basis for the next steps towards realizing TIM, for example for project proposals (note that the planned IPCEI - International Project of Common European Interest - funding of TIM requires individual proposals to be made in each country). Thus, what research projects will ultimately be part of TIM will depend on which project proposals will be submitted and will be successful.

Many of the reviewer comments seemed to be a result of us not being clear enough about this purpose of the paper. We clarified these points in the updated manuscript, for example by discussing what aspects cannot be planned yet, by separating the synergy section into "Summary" and "Preliminary TIM campaign rationale" parts, and by removing details about campaign plans of TIM (like Table 1) because they might still change completely. We also followed most of the detailed suggestions by the reviewer (see responses below).

**General comments:**

I understand that the funding decisions are still very much unknown, and the sources of the funding will dictate (at least in part?) what research topics will be highlighted. However, I highly recommend the following:

1. A lead investigative team needs to be established. It might need to be expanded as additional funding sources are added, but there should be a core team from the start.

Agreed. We currently have emerging target groups on climatology and impacts, idealized cloud modeling, satellite data, radar data, and airborne measurements that are coordinating funding efforts and narrowing down research foci. The leads of these groups will build the core investigative team. This was added to section 3.2.

2. A process needs to be established to identify what research topics will be focused on. Will the lead team decide unilaterally? Will each contributing institution get to make their own decisions? Perhaps there will be a rotation, with focus on, say, hail storms during one period and severe winds the next period.

True, the compartmentalization into institutions, topics, and fixed vs mobile observations will require a clear plan of what to prioritize in what situation. Since each partner organization will be responsible for their funds and local instrument setup, they should have the right to decide about these components. The core team will decide where and when to place intensive, mobile observations. For this, it is important for the core team to be impartial and make decisions on a case-by-case basis informed by the meteorological factors, logistic feasibility, and open discussion with all campaign partners (also see comments on section 2.4 below).

3. It reads like the campaign will be a mix of extended instrumentation deployment coupled with more targeted observing periods. How will these more targeted period be decided upon? Who will be in charge of each one? (While obviously they will want input from everyone, this could easily turn into "too many cooks spoil the soup".)

See above.

4. How will data sharing be handled? Will one organization host everything, or will there just be agreements that the data will be available (and to whom)? What guidelines will be provided for how available the data must be? Who will decide those guidelines?

Part of the funding will be allocated to create a central data hub for systematic data collection, exchange, and long-term archiving. Depending on funding details, such a data hub can be located at one of the partner institutions or at the ESSL. Detailed guidelines will need to be discussed with the community but we plan for the campaign data to be open access for partners. Since research projects are described in individual proposals at an early stage, conflicts about the data use due to similar research topics at later stages seem unlikely. In line with your comment further below, this point was added to section 3.2.

The article needs describe how points 1-4 above will be addressed. If they can't be addressed completely yet (e.g., #3 might depend on who participates), explain how it will be addressed and when. I have a few more targeted comments about this same subject.

● Lines 220- 251 (Section 2.4): This section describes at least 3 different connective modes (supercell, non-supercell pulse storms, MCSs) and 3 hazards (hail, flooding, and downbursts). Each of these represent full lines of research in their own right. It will be difficult to avoid each group trying to optimize observations for their interests at the expense of others. As just one example, observational strategies for the 3 hazards listed above are quite different. Hail needs radar scanning maximized in mid-levels with a focus on high resolution dual pol data. Downbursts need it maximized in low-levels; with the range brought in so velocities aren't aliased.

You are right. Two possible solutions here are to rotate between focus topics or to prioritize depending on the location and forecast. Further planning on this is not yet possible because it

will also depend on what projects will be funded. For example, a much larger research interest in hail than in downbursts can be expected.

● Lines 331-339: The wide-ranging research problems described in this short paragraph (including aerosol impacts on convection! A whole line of research mired in controversy over whether it even exists!) cannot be solved with the 3 new observation types listed. Drones cannot provide in situ microphysical observations within a storm (unless European laws are significantly different than US ones, which of course could be a possibility.) Vertically pointing radars, while helpful, require the thunderstorm to traverse directly overhead, so the sample size will unfortunately be small. While I don't doubt that a targeted set of remote sensing and in situ observations, using existing observation technologies, coupled with carefully structured planned DA and modeling experiments could provide improvement in NWP convective microphysical processes, nothing like that is suggested/described herein. It would need its own dedicated campaign/research effort.

We removed the sentence on aerosols because it is discussed in more detail in section 2.3. We agree with the reviewer that the topic of NWP parameterization improvement is complex. A detailed proposal for how TIM data can be used cannot be put in one paragraph. The current text highlights the importance of this research and that DA experiments could be part of TIM, which we think is enough for this overview paper.

● Figure 5: It seems like almost all of the research topics could fit under both scales and all methods. Can you offer some explanation about why topics are placed under one method/scale and not another? I understand you are still working out the priorities of the project, but even identifying topics that are not to be included would be helpful here. (E.g., data assimilation is mentioned back in Section 2, but doesn't show up Section 3 – is it being saved for later?)

We replaced this figure with a mind-map, highlighting the connections between the different aspects in the subject space.

**Major comments:**

In addition, I have the following major comments:

● Lines 19 - 27: I would not necessarily conclude from Fig. 1 that the largest hail is concentrated near peaks in terrain. Obviously, the southern Alps is a hot spot, but beyond that reports seem fairly scattered. I don't doubt what you say is the case, but I'd recommend including plots of other datasets, like those you mention on lines 23-24, to support your claim, particularly given your main science question in Line 28. Population biases also need to be accounted for.

Further, unless you also want to include global hail data in your intro, I'd change that science question to "why are storms in the vicinity of European mountain ranges more severe"?

While we are leaning against adding additional plots that show pretty much the same thing, especially if they are reproduced from another publication, we see your point. We now refer the reader to Fig. 7, where anyone who remains doubtful can find detailed topography relative to objectively determined report clusters. Since we don't see any way to include all information in one plot (for example different hail sizes) this seemed to be the best solution. We also changed the research question as suggested.

● Lines 163- 165: How will the operations plan handle the mobility challenges that happen near orography? What about forecasting challenges?

ESSL has extensive experience in SCS forecasting and is well-connected with local forecast offices who can contribute their expertise for a given focus region.

Mobility challenges on the other hand, such as crossing mountain ranges or borders, will indeed pose a challenge to TIM. However, it is not necessary to collect observations on every severe weather day of the planned three-year period. Furthermore, in most scenarios severe weather events cluster in time and space, so in our experience it is possible to collect data in the same region over multiple days or by traveling from west to east within a conditionally unstable airmass. This context was added to section 3.2 ("Most severe weather events cluster in time and

space, so mobile teams can collect data in the same region over multiple days or by traveling from west to east within a conditionally unstable airmass…").

● Lines 197- 198: The impact of dust (or even CCN) on convection or CI is still uncertain. I would provide more background information to support this specific idea as a hypothesis.

This paragraph is not referring to the influence of aerosols on microphysics but its impacts on incoming solar radiation. We rephrased the text to make the process clearer and removed the word hypothesis because we are referring to published concepts. We also added the following recent article, which provides further justification for this topic:

https://egusphere.copernicus.org/preprints/2024/egusphere-2024-3924/

In the following paragraph, we also added that the influence of aerosols on SCSs is still uncertain.

● Lines 200-201: Do we have sufficient aerosol measurements (particularly aloft) to make them worth assimilating?

A few point measurements, for example at the existing mountain peak stations like Sonnblick and Jungfraunjoch, could already provide helpful data. At least it would be an interesting modeling experiment. A much more substantial data source could be Lidars or satellite sounders such as on EARTHcare. This was be clarified.

● Lines 273-282: Great points and ideas! However, I'd like to see more explanation about how they will be carried out as they are hard goals to achieve. Are there specific plans to partner with identified educational groups? What about partnerships with the news media (or other typical disseminations of warning information)? What about the forecasters issuing the warnings themselves, are they comfortable incorporating guidance about recommended safety measures/behavior into their warnings? What additional information might they need to provide more tailored recommendations?

ESSL's regular trainings are one route to educate the meteorological community. Another important component is that many students will be drawn to the field of SCS through such a large campaign. More detailed plans to engage with the general public cannot be provided at this stage but we will look into possible connections to the media as suggested. Furthermore, one of the possible funding opportunities for TIM, the "Knowledge for Action in Prevention & Preparedness" call of the European Commision would specifically foster exchange with civil protection agencies:

https://civil-protection-knowledge-network.europa.eu/knowledge-action-prevention-preparedness-2025-call-proposals

Thus, we need to see what funding we get to decide on what outreach possibilities to focus on.

● Lines 340-392: This section has a great description of the many complex problems that can prevent climate models from capturing all the complexities of trends in severe weather. However, one point not explicitly mentioned is the difficulty in translating environmental convective parameters into knowledge of severe weather. Even convection-allowing models that correctly predict the development of thunderstorms still struggle with translating that information into the severe convective hazard itself. I remind the authors that no existing study or method has successfully shown positive skill in forecasting the occurrence of 50 mm hail, for example, and many studies, in fact, have shown no skill at all or even negative skill (e.g., Gange et al. 2017; Adams-Selin et al. 2019, 2023; Gensini et al. 2021). The field campaign plans and increased spatial/temporal observations mentioned in Lines 388-392 may very well uncover new environmental condition - storm dynamics relationships that can improve the situation. However, it is also highly possible that small, stormscale processes may be the differentiator between a storm that produces a severe hazard and one that does not; processes that cannot be identified solely by the surrounding environment (e.g., Adams Selin 2024). If that is the case, it will mean predicting these hazards, and estimating trends in their occurrence

solely from relatively coarse environmental fields, may not be possible. I caution the authors to allow for that result.

You nicely emphasize some of the challenges in translating forecasts to impacts. To us, these give even more motivation for TIM, because studying storm evolution and dynamics and if/how they relate to a given environment is exactly what we need here. We agree that it is possible that these observations will show that storm-internal processes strongly influence hail potential without direct link to environmental changes, which would limit proxy forecasts. Nevertheless, even this would be an important result, which could open new pathways, for example, that more priority is needed for convection allowing models so they can be used more reliably in climate change studies.

We also note that updated environmental proxies seem to show improved skill in hail size prediction compared to older parameters like SHP (e.g., >2 cm and >5 cm), across Europe and the United States, so we don't see this research avenue so pessimistic yet (e.g., Battaglioli et al. 2023a).

● Lines 509-512: Yes, I absolutely agree that data sharing will be one of the key outcomes of this effort. It will be so important, in fact, that I'd like to see more detail here explaining my point #4 above.

See general comment 4.

**Minor comments:**

● Line 26: What makes these population centers highly vulnerable?

Changed to "...are often highly vulnerable because of dense population or specialized agriculture such as wine production. Furthermore…".

● Line 34: Based on Fig. 2, it seems like the survey pre-assumed the severe storms would be near mountains? Again, not a problem as long as that assertion is better motivated up in the intro.

True. We had omitted that point, thanks.

● Lines 53-57: What about LIFT and Swabian MOSES?

We added them, clarifying that they were rather small and localized. Moreover, TIM is benefiting from their expertise, working together in the planning of the campaign.

● Lines 90-91: If you included a lightning rate climatology in section 1, being able to reference the spatial differences between it and OTs and/or hail reports would be a good supporting statement here.

That is a good suggestion, however, we would like to avoid adding additional Figures that are already published elsewhere. Instead, we now provide further reference to Nisi et al. (2018) and Manzato et al. (2022b).

● Line 104: These perturbations would be over a deeper layer than just the boundary layer, right? Boundary layer perturbations alone won't produce the modifications you mention.

Changed to "lower-tropospheric".

● Line 109: "could not be"→ "have not been"

Thanks.

● Line 128-129: Re: upslope few on the leeward side. I agree with the statement, but perhaps include a clarifying phrase about how the upslope flow is opposite to the typical prevailing winds (otherwise, of course, it wouldn't be the leeward side.)

True. Changed it to "...near-ground upslope moisture flux…".

● Lines: 162- 163: Expand upon the results in Feldmann et al. (2023) instead of relying upon Fig. 1.

The part was removed since Feldmann et al. (2023) is discussed elsewhere already.

● Line 210: I'd reference the section where outreach and education efforts will be discussed further.

Good suggestion. Added.

● Lines 221-223: While I agree with your statement here, how frequent are these different types of storms? How difficult will they be to sample, and climatologically, how likely are they?

Around the Swiss Alps, ~75% of radar-classified severe hailstorms do not have a mesocyclone, while ~90% of identified supercells have hail (of any size). So, a considerable fraction of hail storms in the Swiss Prealps is nonsupercellular or at least intermittent, warranting an investigation (Feldmann et al., 2023). We agree that it will be difficult to target single-cell storms not to mention this "special" subset. This should not stop us from at least including the issue in this overview. Again, this article is not a proposal so a full research strategy doesn't seem necessary.

● Lines 258-259: Are tornado surveys planned as part of TIM?

Following up on the development of the IF scale by ESSL, tornado surveys are planned for any strong tornadoes (~IF4+) anyway. Beyond that, TIM will not focus on tornado surveys but with drones likely being used anyway, interesting opportunities may arise.

● Lines 270-272: While comparison of collections of scanned and dissected hailstones between different continents, elevations, etc. are indeed important, they can still only represent a very small percentage of possible hail instances produced by a given storm. Additional observational methods need to be included that will capture larger spatial and temporal areas of the hailswath: e.g., drone survey, time-dependent disdrometer information, a targeted (mobile, if possible) hailpad network.

Agreed. We added that size distributions are also needed. As mentioned in the final section, drone surveys and mobile hail sensors/pads are planned based on the experience of LIFT and ICECHIP.

● Lines 321: Some comparison of the spatial and temporal resolution of currently assimilated observations to the planned TIM observations would be helpful here. Right now the reader doesn't have a good way to assess how impactful these new observations might be. Over how large an area are they planned?

You are perhaps right that the reader lacks information in this section. Unfortunately, the topics of NWP and DA would need their own article to fully capture all the relevant aspects. Thus, we decided that referring to the expert literature, where such information can be found, is enough for the present article. Nevertheless, the section was substantially reworked due to internal discussions, so please check if it is now clearer. In general, several partners in Europe have ongoing or planned projects in Lidar DA, for example using data from Swabian MOSES.
https://www.jstage.jst.go.jp/article/jmsj/98/5/98_2020-049/_article
https://ir.lib.uwo.ca/etd/6356/
https://rmets.onlinelibrary.wiley.com/doi/10.1002/qj.2875

● Lines 326-330: While I agree that more observations are always helpful, my feeling is that determining the location and quantity of such observations to allow for potential improvement of model calibration is not a simple thing. I'd like to see more information from past studies explaining why and how these observations will be configured. Again, one suspects that this goal is worthy of a research project all on its own.

See above.

● Lines 407-409: ESA's new EarthCARE satellite, and the upcoming planned NASA INCUS mission, would both be good satellite datasets to incorporate.

Thanks for the suggestions. In our understanding INCUS will focus on the tropics, no? However, including EarthCARE is a good idea. We will include it and further discuss with our partners at EUMETSAT and DLR.

● Lines 410-422: The described research topics, particularly validation of LI data via ground-based LMAs, will require deployment of a reasonably sized ground network for a not-insignificant amount of time, particularly if measurements of such precision will be made that can be linked to storm microphysics. Will enough of a network be able to be established that these science questions can reasonably be answered?

Colleagues from the Catalunya meteorological service have now joined as TIM partner with their well-established LMA (https://doi.org/10.5194/ems2024-978). We are also in contact with the researchers who set up the LMA used for Relampago to better understand if additional LMAs would be feasible (e.g., https://doi.org/10.1029/2021EA001803). This context was added.

● Lines 428, 432: What is the horizontal and vertical resolution of these new profiling instrument datasets? How much of an impact will these resolutions have on the planned science?

From our understanding, the resolution might still change depending on the scanning strategy and retrieval functions. Please elaborate if you think these are essential to include.

● Lines 442-444: Will these new commercial microwave link networks be established near the campaign location(s) of interest? Where do they monitor? What kind of observations are needed for validation/DA studies?

Commercial microwave link (CML) networks are installed and operated on a country-wide basis by mobile phone network operators. The typical distribution of a modern CML network can be seen in Fig 1 in Blettner et al. 2023 (https://doi.org/10.1029/2023EA002869). In this Figure the data in Germany is only a subset of the available CMLs and limited to one mobile network operator and not including recently installed CMLs (network topology is from the year 2017). In Czech Republic only the western half of the country is included in the available dataset, but the network topology is more recent. From personal communication with mobile network operators we know that the actual CML network in Germany is similarly dense as in Czech Republic and also contains CMLs in the frequency range of 80 GHz. We (KIT) are currently in the process of getting access to CML data for the full current network in Germany. Regarding the last part of the comment, both Ka/ku-band and E-band measurements are suited for data assimilation and first experiments are currently ongoing within the research unit RealPEP funded by DFG.

● Lines 448: What type of airspace regulations re: drones are there in the field campaign location(s) of interest? Will the regulations allow measurements to any reasonable depth? (Drones not being able to fly within thunderstorms due to airspace regulations has been a big frustration of mine.)

Agreed, these are important questions. Proof of concept studies with meteodrones have recently been successful (https://www.mdpi.com/2073-4433/14/9/1382). Whether the general height limitations allow for meaningful data will have to be tested, which is all that can be said at this point. However, we agree about the great potential of drones for within storm data, so we will try to find solutions. For example, collaboration with the military might allow the use of drones in some regions.

● Lines 470-474: Do you have plans for how you will prioritize placement of radars? For example, will filling in gaps in coverage, or establishing the best dual-Doppler coverage be more important? How will those priorities be decided?

Thanks for bringing it up. We think nothing concrete can be decided here until the ultimate projects are clearer.

● Lines 494-495: This sentence encompasses ~5 different lines of research in just the first half of the question!

That is true. We now clarify also in this section that not all topics will be included in the final campaign. We also split up the section into a "summary" and "rationale for TIM" part to make this clearer.

● Line 500: Observations most important to NWP models and forecasting…. For what purpose? Convection, orography, other?

We are not sure if we understand the question? The next sentence makes clear what these observations are needed for. Nowcasting refers to the products described in section 2.6.

● Line 501: What is meant by high resolution? (Rough order of magnitude)

The magnitude is relative to the dataset and application. To be clearer, we changed it to "Higher-resolution datasets (than in operational systems)…".

● Lines 523, 530: What instrumentation will the mobile teams consist of? How will their focus be determined, both overall and IOP-by-IOP? What types of convection will be prioritized?

The sentence now makes clearer that the mobile teams will focus on the instruments described in the previous section ("To sample individual storms and their hazards in such a way, TIM will deploy several mobile teams that can travel to a region where high-impact storms are expected on a given day.") Priorities will be set at a later stage but supercells will likely be the focus because they are the most targetable due to their persistence and they have the largest impact.

● Lines 527-528: Will research topics, in addition to instrumentation, also be based on individual funding?

Yes, this has been made clearer in the new version.

● Lines 532: What kind of climatologies do the ESWD severe report database agree with?

We rephrased this part and some references were adde (e.g., Nisi et al. 2016, Punge & Kunz 2016, Fluck et al. 2021)

● Line 549: How will the numerical modeling experiments be designed and coordinated among groups?

TIM working groups are currently being established to coordinate sub-topics such as numerical modeling experiments. This information was added.

● Line 563: Educating through public outreach and introducing students to convective research are great goals, but no details are provided on activities to achieve those goals.

Good point, we'll need to think about this as TIM goes forward. See major comment 5. Thanks for all your comments.

**Reviewer #2**

Review of EGUsphere-2024-2798

This is an excellent article outlining the arguments for a European field campaign called "Thunderstorm Intensification from Mountains to Plains" (TIM). The paper outlines the high damage potential of severe storms in Europe, summarises the key research topics for TIM to address, and outlines a possible field campaign with focus regions and a synergistic set of instrumentation and research goals. The article is exceptionally well written and compelling. I have only minor comments that should be addressed before the article will be ready to accept. First, references are missing in a few places that I have listed below. Second, while severe storms are affected by climate change and that is a key reason that TIM should go ahead, the language used when writing about the effects of climate change on severe storms needs to be precise about the uncertainties in our current knowledge that exist around certain changes.

We thank the reviewer for the positive feedback and suggestions for improvement. We added the suggested references and context as described below.

**Specific comments**

1. Line 26-27: "are projected to experience the strongest increases in severe weather occurrence as a result of global warming"– I assume this statement relates to projected increases in severe storm environments, which is correct, but since there are possible disconnects between environments and severe weather occurrence (which may also depend on hazard type), care needs to be taken. If the authors mean environments, I would suggest changing this sentence to "increases in atmospheric conditions prone to severe weather occurrence". Also, I think Battaglioli 2023 did not show projection information but rather past trends.

Agreed, we changed the sentence to "… many of these parts of Europe show a strong past or projected increase in atmospheric conditions prone to severe weather occurrence as a result of global warming…"

2. Line 107: "but also the terrain shape and dimensions, and the surface use such as cities or cropland"- the authors should include some references to previous work about these factors.

We removed the words "cities" and "crop land" because they do not fit the context of terrain flows. Houze (2012) fits as a reference for the rest of the sentence.

3. Lines 110-113: "Moisture is often relatively low directly over mountain ridges, leading to less CAPE compared to the surrounding slopes and valleys. Local increase in water vapor content can be found along convergence lines, near lakes, or other regions where moist air can be transported or accumulated, which would lead to larger CAPE, all else being equal." These statements require references.

We added Katona and Markowski 2021, Laiti 2014 and Marquis 2021 from further below in the paper as references and shortened the sentences.

4. Line120: "Nevertheless, studies have suggested"- it is unclear which studies have suggested these findings.

Rephrased to "the studies below have suggested".

5. Line 175: "as well as outflow from existing storms"– any previous work on this in particular?
Will add Soderholm et al. (2014) here from further below.

6. Line 178: "The reliable role of these processes in CI has been shown to increase predictability of convective precipitation over mountains compared to flat terrain"– it is unclear to me what is meant here. Is it that the use of variables related to these processes increases predictability?

Rephrased to "These processes make CI more reliable compared to flat terrain and thus have been shown…".

7. Line 192: "has been proposed"– by who?

We put the references to the end of the sentence and also added the recent De Martin et al. (2025, in press).

8. Line 253: Flash floods should be listed as secondary impacts (since they depend on terrain) and the words "flash floods" in this list could be replaced with "extreme rainfall".

True, thanks.

9. Line 262: "Hail is the dominating source of SCS damage"– I agree, although the ranking could be regionally dependent, and this statement needs a reference.

We removed the sentence since it is not so relevant here and you are right that there is a strong regional dependence.

10.Line 269: The authors should include a reference or two to the numerous wind-driven hail studies that show it is more damaging than non-wind-driven hail.

We added the following reference.

Morgan, G. M., & Towery, N. G. (1976). On the Role of Strong Winds in Damage to Crops by Hail and Its Estimation with a Simple Instrument. *Journal of Applied Meteorology (1962-1982)*, *15*(8), 891–898. Retrieved from http://www.jstor.org/stable/26177518

11.Line 360: "as well as in severe weather reports"– important to include a reference to the study showing this conclusion.

We removed this part as only ESSL reports specifically for hail have been conducted recently.

12.Lines 362-364: "In its Sixth Assessment Report the Intergovernmental Panel on Climate Change (IPCC) states that climate models consistently and with high confidence project environmental changes that would likely lead to an increase in the frequency and intensity of severe thunderstorms, also involving tornadoes, hail, and winds (Calvin et al., 2023, Chapter 11)"– while IPCC gives high confidence to CAPE increases in tropics and subtropics, other projections are given with lower confidence (e.g. there is "significant uncertainty" around projected changes in hail, lightning, and tornadoes.) See e.g. Table 11.2. Given the different trends possible for different hazards (e.g. because hail is affected by melting which can lead to

decreases in surface hail frequency with increased temperature) and given that increases in CAPE do not necessarily always translate into frequency increases (given offsetting effects), the statement made here by the authors should be rephrased to show the level of uncertainty that exists (I note the next paragraph does a good job of mentioning these factors, but the statement still requires slight revision for accuracy).

Agreed, thanks. We removed the "high confidence" part.

13.Line 394: Have advances always been linked to new technology? Would "often" be more accurate?

Yes, that's better.

14. Line 464: A reference for ZDR columns for hail detection is required here.

We added Aregger et al. (2024). https://arxiv.org/abs/2410.10499

**Technical corrections**

Thank you, we corrected these mistakes as suggested.

1. Line 87: "quite long distances" is rather vague and an indication of what it means could be given.

2. Line 152: grammatically, "encounter" should be "encountering" here.

3. Line 196: "fails" should be "fails to occur".

4. The authors have used "exemplary" where I think "example" or "possible" is meant ("an exemplary setup" for example means a gold-standard setup, whereas "an example setup" may be what the authors mean).

5. Line 531: a word (data?) is missing here.

---

## Author Response (AR2)

Author response to round 2 of reviews (minor revisions) of EGUSPHERE-2024-2798
**Invited perspectives: Thunderstorm Intensification from Mountains to Plains**

*We thank the reviewer for taking another thorough look at the manuscript and for these final improvements.*

I thank the authors for taking my suggestions and comments into account. Only minor comments remain (line numbers refer to the tracked changes document):

1. On line 286 the authors say severe convective storms are "now dominating" global insured loss from natural hazards. The report cited shows an earthquake had the highest costs 2023 so the sentence should be rephrased.
*We rephrased the part to "...are one of the dominating drivers of global insured loss from natural catastrophes..."*

2. Line 416: I'm not convinced the evidence for hail is as clear cut as made out here, since increasing CAPE does not always indicate increases in hail at the surface. I would suggest leaving the statement at increases in severe storm environments.
*We removed the specific mention of hail. It now reads:*
*"In its Sixth Assessment Report the Intergovernmental Panel on Climate Change (IPCC) states that climate models consistently project environmental changes that would likely lead to an increase in the frequency and intensity of severe thunderstorms."*

3. Line 418: Suggest "One reason" is replaced with "One reason for the high uncertainty".
*Done*

4. Line 501: EarthCare requires a reference.
*Agreed, we added the summary of Wehr et al. (2023).*
*Wehr, T., Kubota, T., Tzeremes, G., Wallace, K., Nakatsuka, H., Ohno, Y., … Bernaerts, D. (2023). The EarthCARE mission - science and system overview. Atmospheric Measurement Techniques, 16(15), 3581–3608. https://doi.org/10.5194/amt-16-3581-2023*

5. Figure 5: The different shadings of grey should be explained in the caption. For example, I don't know why broader applications are listed within a box but reanalysis and km-scale modelling get their own boxes.
*We clarified that the dark grey boxes correspond to the research subjects and the light grey ones to datasets. We also split the grey boxes to be consistent.*

6. The list of research questions starting around line 570 includes research topic numbers (RTs) from 1 to 8, whereas the topic descriptions are numbered from 2.1 to 2.8. For consistency the numbering should be the same.
*Agreed, thanks.*

7. Line 615: This sentence seems a little lost and should be added to a paragraph. Same for the sentences on lines 643 and 656.
*Thanks for pointing out this formatting error. We deleted the line break.*

8. Line 626 and following: "ROIs" should include an s when referred to in the plural, for clarity.
*Added.*

9. Line 650: Remove unnecessary capitalisation on climatology and radar.
*Done, thank you.*